# Interactions between calmodulin and neurogranin govern the dynamics of CaMKII as a leaky integrator

**Mariam Ordyan**[1,2], **Tom Bartol**[2], **Mary Kennedy**[3], **Padmini Rangamani**[4]*, **Terrence Sejnowski**[1,2]*

**1** Institute for Neural Computation, University of California San Diego, La Jolla, California, United States of America, **2** Computational Neurobiology Laboratory, Salk Institute for Biological Sciences, La Jolla, California, United States of America, **3** The Division of Biology and Biological Engineering, California Institute of Technology, Pasadena, California, United States of America, **4** Department of Mechanical and Aerospace Engineering, University of California San Diego, La Jolla, California, United States of America

* prangamani@ucsd.edu (PR), terry@salk.edu (TS)

**Data Availability Statement:** All bngl files are available on GitHub at the following repository:

## Abstract

Calmodulin-dependent kinase II (CaMKII) has long been known to play an important role in learning and memory as well as long term potentiation (LTP). More recently it has been suggested that it might be involved in the time averaging of synaptic signals, which can then lead to the high precision of information stored at a single synapse. However, the role of the scaffolding molecule, neurogranin (Ng), in governing the dynamics of CaMKII is not yet fully understood. In this work, we adopt a rule-based modeling approach through the Monte Carlo method to study the effect of $Ca^{2+}$ signals on the dynamics of CaMKII phosphorylation in the postsynaptic density (PSD). Calcium surges are observed in synaptic spines during an EPSP and back-propagating action potential due to the opening of NMDA receptors and voltage dependent calcium channels. Using agent-based models, we computationally investigate the dynamics of phosphorylation of CaMKII monomers and dodecameric holoenzymes. The scaffolding molecule, Ng, when present in significant concentration, limits the availability of free calmodulin (CaM), the protein which activates CaMKII in the presence of calcium. We show that Ng plays an important modulatory role in CaMKII phosphorylation following a surge of high calcium concentration. We find a non-intuitive dependence of this effect on CaM concentration that results from the different affinities of CaM for CaMKII depending on the number of calcium ions bound to the former. It has been shown previously that in the absence of phosphatase, CaMKII monomers integrate over $Ca^{2+}$ signals of certain frequencies through autophosphorylation (Pepke et al, Plos Comp. Bio., 2010). We also study the effect of multiple calcium spikes on CaMKII holoenzyme autophosphorylation, and show that in the presence of phosphatase, CaMKII behaves as a leaky integrator of calcium signals, a result that has been recently observed *in vivo*. Our models predict that the parameters of this leaky integrator are finely tuned through the interactions of Ng, CaM, CaMKII, and PP1, providing a mechanism to precisely control the sensitivity of synapses to calcium signals. Author Summary not valid for PLOS ONE submissions.

https://github.com/marordyan/CaMKII_well_mixed/.

**Funding:** PR,MO,TB,TS: FA9550-18-1-0051 US Air Force https://www.airforce.com/, TB,TS: P41GM103712 National Institute of Health https://www.nih.gov/ MK,TB,TS: NS44306MK National Institute of Health https://www.nih.gov/ MK,TB,TS: DA030749 National Institute of Health https://www.nih.gov/ The funders had no role in study design, data collection and analysis, decision to publish, or preparation of the manuscript.

**Competing interests:** The authors have declared that no competing interests exist.

## Author summary

Neurons communicate with each other through synapses. The strength of a particular synapse is effectively the level of sensitivity of the postsynaptic neuron in response to firing of the presynaptic neuron. The process of changing synaptic strength is dubbed synaptic plasticity, a foundational aspect of learning and memory. In this work, we create a computational model of a calcium signaling pathway that sets off a chain reaction in CaMKII phosphorylation, eventually leading to synaptic plasticity. Computational modeling provides a unique way to tease apart and understand the non-intuitive results of interactions between the molecules involved. Our model successfully predicts the experimentally observed activation dynamics of this crucially important enzyme which is necessary for learning. These dynamics, along with other pathways, regulate the size of the synapse, which is known to be highly correlated with synaptic strength. In this work, we reveal quantitative characteristics of CaMKII activation for various stimuli, leading to important insights regarding the potential role of Neurogranin, a scaffolding protein in this pathway.

## Introduction

Information is stored in the brain through synaptic plasticity. It has been reported that synaptic strength is highly correlated with the size of the spine head, and the precision of information stored at a single synapse is quite high despite the stochastic variability of synaptic activation [1–3]. Structural changes to the postsynaptic spine that can lead to spine enlargement, and thus structural plasticity are triggered by $Ca^{2+}$ signaling [4, 5]. Time averaging of these calcium signals has been suggested as a plausible mechanism for achieving the high precision of information processing observed in spines. Furthermore, phosphorylation of calcium/calmodulin-dependent protein kinase II (CaMKII) has been postulated as the most probable pathway satisfying the long time scales predicted for averaging [1].

CaMKII is an autophosphorylating kinase [6, 7] [8–10]; in postsynaptic densities (PSD), CaMKII has been shown to play an important role in learning and memory [11]. Specifically, mice with a mutation in a subtype of CaMKII exhibit deficiencies in long-term potentiation (LTP) and spatial learning [10, 12, 13]. Moreover, CaMKII expression regulates the rate of dendritic arborization and the branch dynamics [14, 15], highlighting its importance in structural plasticity. Additionally, CaMKII has been shown to bind actin, the major cytoskeletal protein in dendritic spines [16–18], further emphasizing its role in structural plasticity. Activation of CaMKII is exquisitely regulated at multiple levels as summarized below.

- **CaMKII activation by calmodulin (CaM)**: CaMKII is activated by calmodulin (CaM) [6], which is a protein with 4 $Ca^{2+}$ binding sites: 2 on its C-terminal and 2 on N-terminal (Fig 1A) [19, 20]. CaM binds $Ca^{2+}$ cooperatively and is able to activate CaMKII more potently if it has more $Ca^{2+}$ ions bound to it [21].

- **Neurogranin (Ng)-CaM interactions**: In the absence of $Ca^{2+}$, CaM is bound to scaffolding protein neurogranin (Ng), which dramatically reduces the affinity of CaM for $Ca^{2+}$ (Fig 1B). On the other hand, $Ca^{2+}$ decreases binding affinity of CaM for Ng [22]. Thus, CaM activation and therefore CaMKII activation depend on the competitive effects of $Ca^{2+}$ and Ng.

- **The role of structure in CaMKII activation**: Further complexity for CaMKII activation is built into the structure of the molecule itself. CaMKII is a dodecamer arranged in 2 stacked hexomeric rings [6, 23]. Monomers of CaMKII truncated to remove the association domain

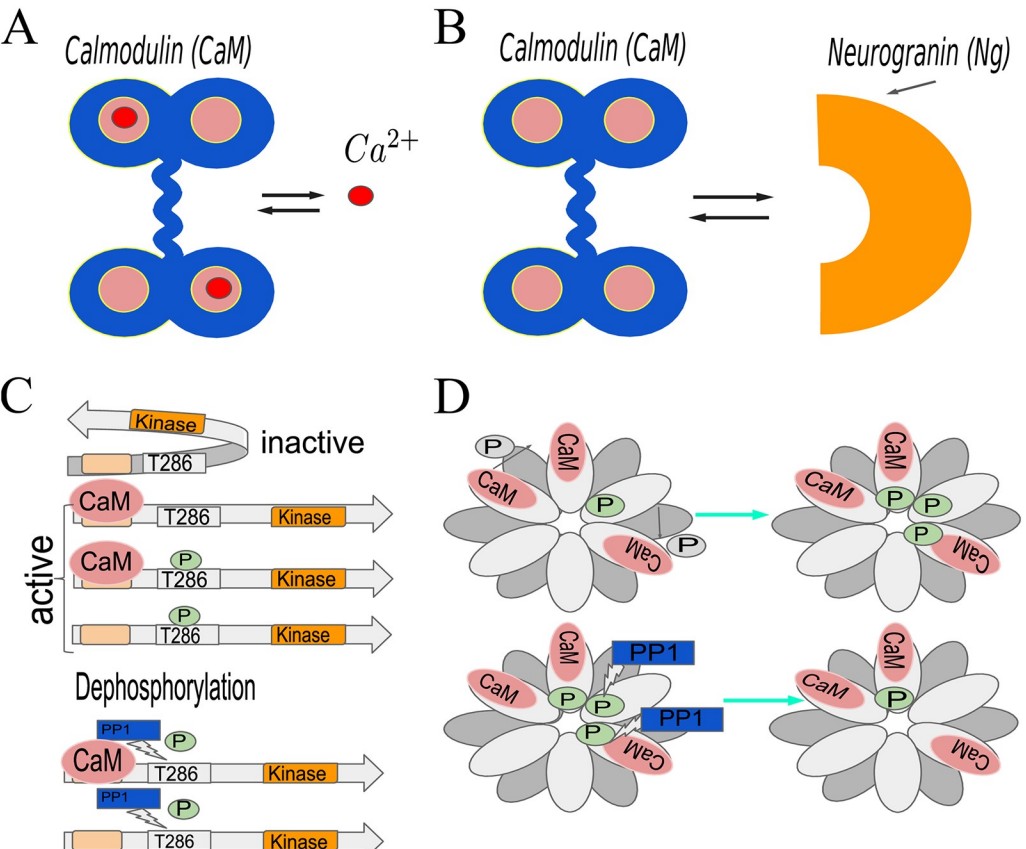

**Fig 1. Schematic representation of the interactions between calmodulin (CaM), calcium, CaMKII, and neurogranin (Ng).** (A) Calmodulin has 4 calcium binding sites; it binds $Ca^{2+}$ cooperatively and its ability to activate CaMKII depends on how many of these sites are occupied. (B) Neurogranin (Ng) is a scaffolding molecule; upon binding calmodulin it dramatically reduces the latter's affinity for calcium. In this work, we assume that CaM cannot bind calcium and Ng simultaneously. (C) The default conformation of CaMKII monomers is inactive (top); they can be activated by CaM binding. Once bound to CaM, the monomers can be phosphorylated by another active CaMKII protein. In this case, the CaMKII monomer will remain active even after losing CaM. Protein phosphatase 1 (PP1) dephosphorylates CaMKII (bottom). (D) CaMKII holoenzyme is a dodecamer, which consists of 2 stacked hexomeric rings of CaMKII monomers. Within the ring, a given CaMKII monomer can be phosphorylated by its neighbor provided that they are both in the active conformation. This is commonly referred to as autophosphorylation.

(mCaMKII) contain a kinase domain, CaM-binding domain, and phosphorylation sites T286 (287), and T305(306) [9, 24–26]. When the CaM-binding domain is unoccupied, and the T286(287) site is unphosphorylated, the monomer is in a conformation such that the kinase is inactive (Fig 1C, top). When CaM is bound to a CaMKII monomer, the latter undergoes a conformational change, such that the kinase is now active. An active CaMKII monomer can phosphorylate other $Ca^{2+}$/$CaM$-bound CaMKII monomers resulting in CaMKII autophosphorylation. A CaMKII monomer can only be phosphorylated if the calmodulin bound to it has at least one calcium ion, and once phosphorylated [27], the monomer remains in the active state even after unbinding from CaM (Fig 1C middle). The same is true for full length monomers that are bound within the holoenzyme. In this manner, a brief $Ca^{2+}$ influx initiates a prolonged CaMKII activation [7, 9, 26, 27, 27–29]. Within the holoenzyme each of the monomers can phosphorylate neighbors, as long as the former is active and the latter has CaM bound to it (Fig 1D) [8, 27]. The phosphorylation rate of the CaMKII monomers depends on how many $Ca^{2+}$ ions are bound to the CaM bound to that substrate [30].

- **Phosphatases are important**: Various types of protein phosphatases (PP1, PP2A, PP2B, PP2C) can dephosphorylate CaMKII in the brain and, if CaM is not bound to the latter, bring it back to its inactive state [25, 31]. In the PSD, however, protein phosphatase 1 (PP1) has been shown to be mainly responsible for CaMKII dephosphorylation [31, 32] (Fig 1C bottom).

Many computational models of CaMKII dynamics have been developed in the literature to probe the different interactions in this cascade at varying levels of detail [11, 33–40]. A large majority of these models focused on the bistability behavior of CaMKII [11, 33, 35, 37, 38, 40], which resulted from the nonlinear rate functions used in the model. However, experiments suggest that in long-term potentiation, CaMKII is only transiently activated and does not exhibit bistable behavior [41, 42]. Furthermore, as noted recently [43], the behavior of CaMKII is complicated by its multimeric nature and by the intersubunit interaction necessary for phosphorylation of T286, which is affected by the presence of Ng and protein phosphatases. Therefore, a more complete computational model of CaMKII dynamics needs to account for both the behavior of the monomers and the dynamics of CaMKII holoenzyme in the presence of Ng and PP1.

Here we sought to examine how the competition between $Ca^{2+}$-mediated activation of CaM and Ng scaffolding of CaM affects the response of CaMKII to calcium signals. To do so, we developed two computational models—Model 1 that accounts for CaMKII monomer activation and Model 2 that investigates holoenzyme kinetics—in an agent-based framework. Model 1 is built on a previously published model of activation of CaMKII monomers by $Ca^{2+}/CaM$ [30] and now includes the role of the scaffolding molecule Ng and the protein phosphatase PP1. Using these models we investigated the dynamics of monomeric and dodecameric CaMKII phosphorylation as a function of the dynamics of $Ca^{2+}$-influx and of interactions with Ng. An important distinction between our model and those presented in [34] and [44] is that we did not explicitly construct our model to replicate desired/observed CaMKII activation dynamics. Rather, our model hinges solely on rate constants for interactions between molecules based on biochemical experiments and presented in Tables 1 and 2. Our results show that under conditions of our model CaMKII behaves as a leaky integrator and, more importantly, that the scaffold molecule, Ng, tunes the behavior of this leaky integrator.

## Methods

We constructed the models at different scales to characterize CaMKII phosphorylation at increasing levels of complexity. First, we added the scaffolding molecule Ng to the model from *Pepke et al* [30], to investigate the effect of Ng on CaMKII phosphorylation dynamics. Second, we added PP1 to this CaMKII monomers model to simulate the phosphorylation-dephosphorylation cycle and characterize the effects of Ng on this system. Finally, we built a model of a dodecameric holoenzyme and looked at the response of the holoenzyme-phosphatase system to calcium signals and how it is affected by Ng. An important assumption of all the models developed in this study is that only one of the phosphorylation sites (T286/7) of CaMKII is considered throughout. The phosphorylation of T305/6 site is a slower reaction that is known to inhibit CaM binding to T286/7-unphosphorylated CaMKII subunit [114], and is omitted from our simulations.

### Model description

**Model of CaMKII monomers.** We begin with the model of CaMKII monomers (mCaMKII) described by Pepke et al. [30]. This model includes $Ca^{2+}$ binding rates to CaM, CaM

**Table 1. Reaction rates for the model.**

| Description | Parameter | Value | Reference | Parameter | Value | Reference |
|---|---|---|---|---|---|---|
| $Ca^{2+}$ binding to CaM | $k_{on}^{1C}$ | 4 $\mu M^{-1}s^{-1}$ | [21, 30, 45, 46] | $k_{off}^{1C}$ | 40.24 $s^{-1}$ | [47–49] |
| | $k_{on}^{2C}$ | 10 $\mu M^{-1}s^{-1}$ | [21, 30, 45, 46] | $k_{off}^{2C}$ | 9.3 $s^{-1}$ | [49] |
| | $k_{on}^{1N}$ | 100 $\mu M^{-1}s^{-1}$ | [21, 30, 45, 46] | $k_{off}^{1N}$ | 2660 $s^{-1}$ | [47, 49–51] |
| | $k_{on}^{2N}$ | 150 $\mu M^{-1}s^{-1}$ | [21, 30, 45, 46] | $k_{off}^{2N}$ | 990 $s^{-1}$ | [47–51] |
| CaM binding to unphosphorylated CaMKII | $k_{on}^{CaM0}$ | $3.8 \cdot 10^{-3}$ $M^{-1}s^{-1}$ | [30, 52] | $k_{off}^{CaM0}$ | 6.56 $s^{-1}$ | [30, 52] |
| | $k_{on}^{CaM1C}$ | $59 \cdot 10^{-3}$ $\mu M^{-1}s^{-1}$ | [30, 52] | $k_{off}^{CaM1C}$ | 6.72 $s^{-1}$ | [30, 52] |
| | $k_{on}^{CaM2C}$ | 0.92 $\mu M^{-1}s^{-1}$ | [30, 52] | $k_{off}^{CaM2C}$ | 6.35 $s^{-1}$ | [30, 52] |
| | $k_{on}^{CaM1C1N}$ | 0.33 $\mu M^{-1}s^{-1}$ | [30, 52] | $k_{off}^{CaM1C1N}$ | 5.68 $s^{-1}$ | [30, 52] |
| | $k_{on}^{CaM2C1N}$ | 5.2 $\mu M^{-1}s^{-1}$ | [30, 52] | $k_{off}^{CaM2C1N}$ | 5.25 $s^{-1}$ | [30, 52] |
| | $k_{on}^{CaM1N}$ | $22 \cdot 10^{-3}$ $\mu M^{-1}s^{-1}$ | [30, 52] | $k_{off}^{CaM1N}$ | 5.75 $s^{-1}$ | [30, 52] |
| | $k_{on}^{CaM2N}$ | 0.1 $\mu M^{-1}s^{-1}$ | [30, 52] | $k_{off}^{CaM2N}$ | 1.68 $s^{-1}$ | [30, 52] |
| | $k_{on}^{CaM1C2N}$ | 1.9 $\mu M^{-1}s^{-1}$ | [30, 52] | $k_{off}^{CaM1C2N}$ | 2.09 $s^{-1}$ | [30] |
| | $k_{on}^{CaM4}$ | 30 $\mu M^{-1}s^{-1}$ | [30, 52] | $k_{off}^{CaM4}$ | 1.95 $s^{-1}$ | [30, 52] |
| $Ca^{2+}$ binding to CaM-CaMKII | $k_{on}^{K1C}$ | 44 $\mu M^{-1}s^{-1}$ | [30, 49] | $k_{off}^{K1C}$ | 29.04 $s^{-1}$ | [30, 49] |
| | $k_{on}^{K2C}$ | 44 $\mu M^{-1}s^{-1}$ | [30, 49] | $k_{off}^{K2C}$ | 2.42 $s^{-1}$ | [30, 49] |
| | $k_{on}^{K1N}$ | 75 $\mu M^{-1}s^{-1}$ | [30, 49] | $k_{off}^{K1N}$ | 301.5 $s^{-1}$ | [30, 49] |
| | $k_{on}^{K2N}$ | 76 $\mu M^{-1}s^{-1}$ | [30, 49] | $k_{off}^{K2N}$ | 32.68 $s^{-1}$ | [30, 49] |
| CaMKII binding to CaM-CaMKII * | $k_{on}^{CaMKII}$ | 50 $\mu M^{-1}s^{-1}$ | [30, 53] | $k_{off}^{CaMKII}$ | 60 $s^{-1}$ | [27, 30, 54] |
| CaMKII phosphorylation | $k_{p}^{CaM1C}$ | 0.032 $s^{-1}$ | [21, 30] | $k_{p}^{CaM2C}$ | 0.064 $s^{-1}$ | [21, 30] |
| | $k_{p}^{CaM1C1N}$ | 0.094 $s^{-1}$ | [21, 30] | $k_{p}^{CaM2C1N}$ | 0.124 $s^{-1}$ | [21, 30] |
| | $k_{p}^{CaM1N}$ | 0.061 $s^{-1}$ | [21, 30] | $k_{p}^{CaM2N}$ | 0.12 $s^{-1}$ | [21, 30] |
| | $k_{p}^{CaM1C2N}$ | 0.154 $s^{-1}$ | [21, 30] | $k_{p}^{CaM4}$ | 0.96 $s^{-1}$ | [21, 30] |
| CaM binding to phosphorylated CaMKII | $k_{p,on}^{CaM0}$ | $1.27 \cdot 10^{-3}$ $\mu M^{-1}s^{-1}$ | [30, 52] | $k_{p,on}^{CaM1C}$ | 19.7 $\mu M^{-1}s^{-1}$ | [30, 52] |
| | $k_{p,on}^{CaM2C}$ | 0.3 $\mu M^{-1}s^{-1}$ | [30, 52] | $k_{p,on}^{CaM1C1N}$ | 1.1 $\mu M^{-1}s^{-1}$ | [30, 52] |
| | $k_{p,on}^{CaM2C1N}$ | 1.73 $\mu M^{-1}s^{-1}$ | [30, 52] | $k_{p,on}^{CaM1N}$ | 7.3 $\mu M^{-1}s^{-1}$ | [30, 52] |
| | $k_{p,on}^{CaM2N}$ | 0.03 $\mu M^{-1}s^{-1}$ | [30, 52] | $k_{p,on}^{CaM1C2N}$ | 0.63 $\mu M^{-1}s^{-1}$ | [30, 52] |
| | $k_{p,on}^{CaM4}$ | 10 $\mu M^{-1}s^{-1}$ | [30, 52] | $k_{p,off}^{CaM}$ | 0.07 $s^{-1}$ | [52] |
| Ng binding to CaM | $k_{on}^{Ng}$ | 5 $\mu M^{-1}s^{-1}$ | [55] | $k_{off}^{Ng}$ | 1 $s^{-1}$ | [55] |
| CaMKII dephosphorylation by PP1 (Michaelis-Menten constants) | $k_{cat}$ | 0.41 $s^{-1}$ | [56] | $K_m$ | 11 $\mu M$ | [56] |

All the numbers with the exception of "CaMKII binding to CaM-CaMKII" are used in both monomers and holoenzyme models. The difference between the 2 models are the initial conditions: we either start the simulations with only monomers or only holoenzyme, and do not allow the disintegration of the latter.

* only present in the model for monomers

binding rates to mCaMKII, and phosphorylation rates of active CaMKII monomers by one another, all depending on how many $Ca^{2+}$ ions are bound to the CaM molecules involved. We incorporate Ng binding to CaM with the rate constants from [55], and assume that the binding of CaM to $Ca^{2+}$ and Ng is mutually exclusive (Table 1). In addition to the reactions in [30], we included CaM unbinding and binding to phosphorylated CaMKII, albeit with slower kinetics [52]. This reaction is important for the timescales of our interest (on the order of minutes) and we adapt these reaction rates from [52]. Finally, CaMKII dephosphorylation by PP1 is modeled as Michaelis-Menten kinetics, with the rate constants from [56] (Table 1).

**Table 2. Reactions implemented in our models.**

$$Ca^{2+} + CaM((n-1)Ca^{2+}) \underset{k_{off}^{n}}{\overset{k_{on}^{n}}{\rightleftharpoons}} CaM(nCa^{2+})$$

$$Ng + CaM(0Ca^{2+}) \underset{k_{off}^{ng}}{\overset{k_{on}^{ng}}{\rightleftharpoons}} CaM \cdot Ng$$

$$CaMKII + CaM(nCa^{2+}) \underset{k_{off}^{CaMnCa}}{\overset{k_{on}^{CaMnCa}}{\rightleftharpoons}} CaMKII \cdot CaM(nCa^{2+})$$

$$CaMKII \cdot CaM((n-1)Ca^{2+}) + Ca^{2+} \underset{k_{off}^{KnCa}}{\overset{k_{on}^{KnCa}}{\rightleftharpoons}} CaMKII \cdot CaM(nCa^{2+})$$

$$CaMKII \cdot CaM(nCa^{2+}) + CaMKII \cdot CaM(mCa^{2+}) \underset{k_{off}^{CaMKII}}{\overset{k_{on}^{CaMKII}}{\rightleftharpoons}} CaMKII \cdot CaM(nCa^{2+}) \cdot CaMKII \cdot CaM(mCa^{2+})$$

$$CaMKII \cdot CaM(nCa^{2+}) \cdot CaMKII \cdot CaM(mCa^{2+}) \overset{k_{p}^{CaMmCa}}{\longrightarrow} CaMKII \cdot CaM(nCa^{2+}) \cdot pCaMKII \cdot CaM(mCa^{2+})$$

$$pCaMKII \cdot CaM(nCa^{2+}) \underset{k_{p,on}^{CaMnCa}}{\overset{k_{p,off}^{CaM}}{\rightleftharpoons}} pCaMKII + CaM(nCa^{2+})$$

$$PP_1 + pCaMKII \cdot CaM(nCa^{2+}) \overset{k_M}{\rightleftharpoons} PP_1 \cdot pCaMKII \cdot CaM(nCa^{2+}) \overset{k_{cat}}{\longrightarrow} PP_1 + CaMKII \cdot CaM(nCa^{2+})$$

Listed are the reactions implemented in both models with the exception of the 5th reaction between 2 CaMKII monomers, that is only present in the mCaMKII model since in the hCaMKII model the the subunits are already linked to one another within the holoenzyme. Here n stands for the number and different configurations of $Ca^{2+}$ ions bound to CaM (thus $nCa^{2+}$ should be read as 'aCbN', where a and b can range from 0 to 2, see Table 1.)

**Model of CaMKII holoenzyme.** **Assumptions specific to the holoenzyme model**: It has been shown that while the kinase domains of individual subunits are attached to the rigid hub domain with a highly flexible linker domain, $\sim$20% of subunits form dimers, and $\sim$ 3% of them are in a compact conformation, both of which render CaM-binding-domain inaccessible [115]. Here, we do not include such detailed interactions between linker domains and flexibility of the individual subunits within the holoenzyme. Rather, we assume that the kinase domains are positioned rigidly within the 2 hexameric rings, such that each subunit is in a position to phosphorylate only one of its neighbors as depicted in Fig 1D.

We further assume that the CaMKII subunits within the holoenzyme have the same binding rates to different species of $Ca^{2+}/CaM$ as the CaMKII monomers, and once activated their phosphorylation rate is the same as that of the corresponding monomers. Any possible

allosteric interactions are ignored. Based on these assumptions, the model for the CaMKII holoenzyme does not include the reaction for 2 CaMKII monomers binding to one another. Rather, once the appropriate neighbor of an active CaMKII subunit binds $Ca^{2+}/CaM$, the phosphorylation rule is invoked with the corresponding reaction rate (Table 1).

**Model development.** To model the dynamics of CaMKII holoenzyme, we changed the types of molecules that we start the simulation with. Instead of CaMKII monomers, we initiated the simulation with CaMKII holoenzymes, and defined the phosphorylation rules for individual subunits such that a given subunit can only phosphorylate one of its neighbors (see model assumptions). The rules are the same as those of the monomers, with the exception of 2 CaMKII monomers binding each other to phosphorylate one another, since they are already bound within the holoenzyme. In this case, the reaction rules apply to the CaMKII subunits within the holoenzyme rather than CaMKII monomers.

This model does not contain any CaMKII monomers outside of the holoenzyme. We calculated the total number of the holoenzymes to keep the concentration of the $[mCaMKII] = 80\ \mu M$ consistent with our model of the monomers. For our simulated PSD volume $V = 0.0156\ \mu m^3$ this concentration corresponds to $80\ \mu M = 80 \cdot (6.022 \cdot 10^{23} \cdot 0.0156 \cdot 10^{-15}) \cdot 10^{-6}$ monomers which constitutes $752/12 \approx 63$ holoenzymes.

## Rule-based modeling and BioNetGen

Since each CaMKII monomer can bind a calmodulin molecule, which can adopt 9 distinct states (assuming that the 2 $Ca^{2+}$ binding sites on each of the lobes are indistinguishable), and be phosphorylated or unphosphorylated, there are more than $18^{12}$ states the dodecameric holoenzyme could adopt. To avoid a combinatorial explosion of states we built our model in a rule-based modeling language BioNetGen [116], which is briefly described here.

A powerful idea at the heart of rule-based modeling is colloquially referred as "don't care, don't write" [117, 118]. Here, for a given reaction (or rule), we only specify the states of the reactants that are relevant for the said reaction, and leave the rest unspecified. For example, a specific CaMKII subunit will bind a given species of $Ca^{2+}/CaM$ with the same rate regardless of the conformation of its neighbors. Thus, a rule to bind a CaM molecule that has 2 $Ca^{2+}$ ions bound to its C-terminus would look like:

$CaMKII(286 \sim U, cam) + CaM(camkii, C \sim 2, N \sim 0) \Longleftrightarrow CaMKII(286 \sim U, cam!1).CaM(camkii!1, C \sim 2, N \sim 0), k_{on}^{CaM2C},\ k_{off}^{CaM2C}$.

This rule indicates that regardless of the presence or conformations of a non-phosphorylated (indicated by $286 \sim U$) CaMKII subunits' neighbors it binds CaM, with 2 $Ca^{2+}$ ions at its C lobe, reversibly with the reaction rates $k_{on}^{CaM2C}$ and $k_{off}^{CaM2C}$ (Table 1). This dramatically reduces the number of reactions that need to be written. In the case of our CaMKII model, we only end up with $\sim 40$ reactions despite the huge number of possible states.

In BioNetGen, once the reactions have been specified, if there is no danger of running into combinatorial explosion (the numbers of possible states and reactions are manageable), a full network of reactions can be generated, and the simulations can be run with ordinary differential equations (ODEs) [118]. If the number of states of a given enzyme is too large however, a stochastic agent-based approach is adopted, in which case only the reactions between existing discrete molecules that occur during the simulation need to be tracked by the program (network free simulation [119, 120]). In this case, the important quantity is the number of different states present which can be much smaller than the number of possible states. In this work, we conduct the simulations for the monomer model with ODEs, and for the holoenzyme model with the network free method.

### Modeling calcium inputs

To implement $Ca^{2+}$ pulses, we needed to include time-dependent rate constants into our model. For this purpose we concocted the following reaction:

$$0 \Rightarrow t(), ticspersecond$$

This allowed us to implement a reaction with a rate constant dependent on t(), namely a $Ca^{2+}$ pulse:

$$0 \Rightarrow Ca, \ \ A * (t()/\tau_R) * e^{-(t()/\tau_F)}$$

where A, $\tau_R$, and $\tau_F$ are the amplitude, and the raising and falling time constants of the reaction rate (see the model). These parameters where chosen such as to produces the desired shapes of the $Ca^{2+}_{free}$ pulses.

## Results

Distribution of different calcium-bound CaM depends on the presence or absence of Ng. We assumed that all molecules were well mixed and adopted a rule-based Monte Carlo approach to simulate the reaction dynamics (see Methods). Since Ng is a scaffolding molecule and its presence is expected to decrease the $[CaM]_{free}$: $[Ca^{2+}]_{free}$ ratio we first investigated how it affects the distribution of different calcium-bound CaM molecules while maintaining $[Ca^{2+}]$ constant. Following Pepke *et al.* [30], we set $[mCaMKII]$ = 80 µM and calculated the dose response to calmodulin at a constant calcium concentration of $[Ca^{2+}]$ = 10 µM with and without Ng. When present, the concentration of Ng was set to 20 µM [34].

Fig 2 shows the maximum amounts of each individual $Ca^{2+}/CaM$ species relative to total CaM available for a range of [CaM]s. We observe that without Ng, we consistently have more $1Ca^{2+}/CaM$ than in the presence of Ng (Fig 2A, 2B and 2E top panels). This is easily understandable: Ng limits the availability of free calmodulin, and in its absence, we have more CaM available to bind calcium. On the other hand, when we look at CaM species that have multiple calcium ions bound to them, we see a crossover of the 2 curves as the total [CaM] increases. This is particularly striking for $3Ca^{2+}/CaM$ and $4Ca^{2+}/CaM$ (Fig 2C, 2D and 2E bottom panels). For these species, limiting the available free CaM can be beneficial since this increases the ratio $[Ca^{2+}]$:$[CaM]_{free}$, resulting in higher number of calmodulin proteins bound to multiple calcium ions. Fig 2E shows the dose response in the physiological range of CaM concentration.

While the calculated effects may seem small, they are amplified by the fact that these multi-calcium-bound calmodulin species have a higher CaMKII binding rate, and render the bound CaMKII more susceptible to phosphorylation (Table 1). These observations lead to an interesting question: is the higher relative concentration of $4Ca^{2+}/CaM$ resulting truly from the competition between calcium and Ng for CaM or does CaMKII play a role as a leaky buffer for different multi-calcium-bound forms? To answer this question, we repeated the simulations in the absence of any CaMKII. In this case, we observed an even more pronounced effect of Ng in the increase of multi-calcium-bound states of CaM for certain CaM concentrations (S1 Fig) but the overall trend remains the same as in the presence of CaMKII (Fig 2). Thus, we conclude that the relative distribution of different calcium-bound forms of CaM are indeed a result of competition between Ng and $Ca^{2+}$ for CaM.

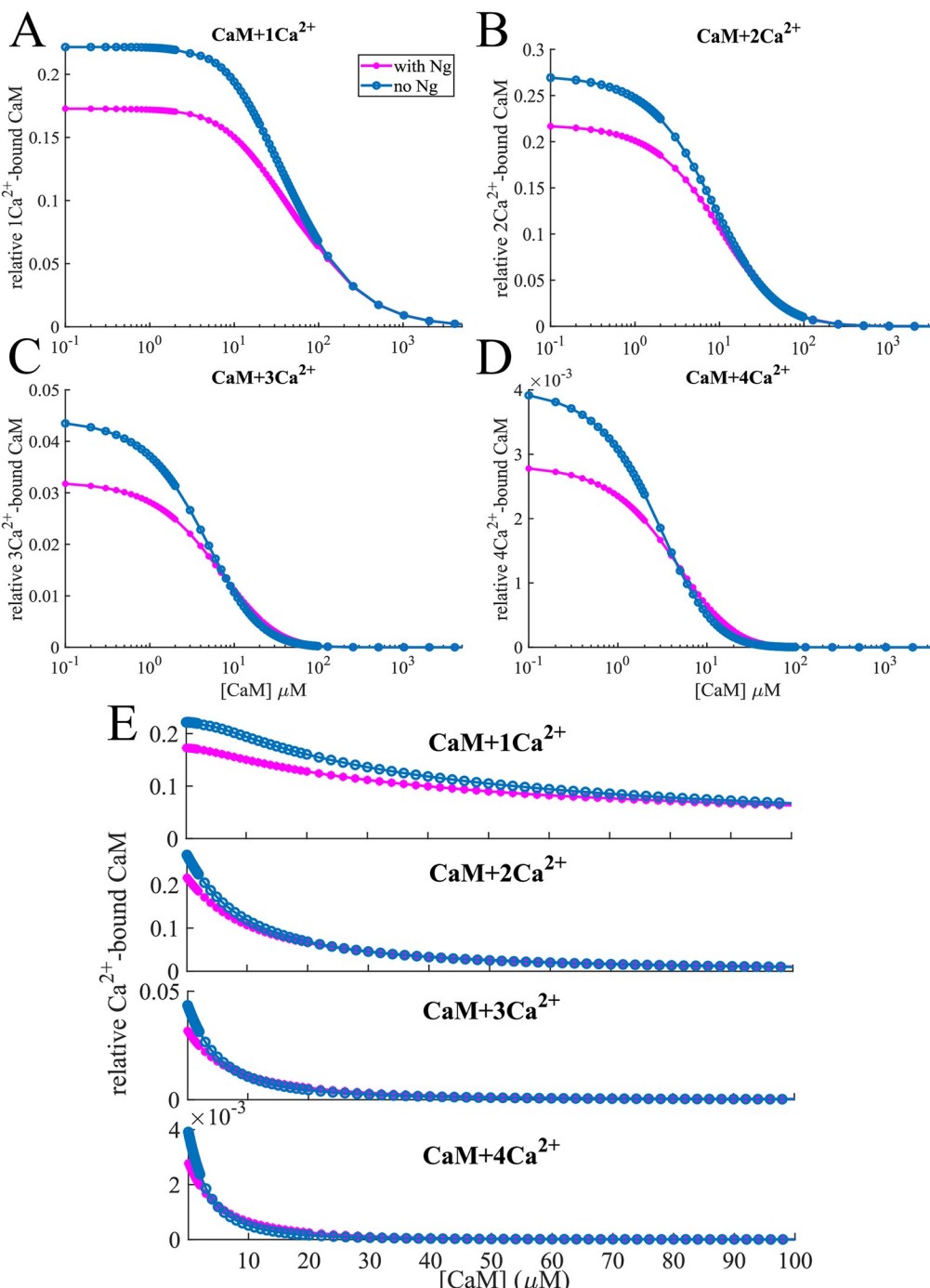

**Fig 2. CaM binding calcium.** Relative concentration of (A) 1-, (B) 2-, (C) 3- and (D) 4-calcium-bound calmodulin vs. total [CaM] at constant $[Ca^{2+}]$ = 10 μM. For most of the calcium-bound forms, the relative concentration is lower in the presence of neurogranin. However, in the case of 3- and 4-calcium-bound calmodulin the situation is reversed for certain [CaM]: Ng competes with $Ca^{2+}$ for free calmodulin, which allows a regime where the concentration of these "multi-calcium-bound" forms is higher. As the concentration of calmodulin is further increased, the $[Ca^{2+}]$:$[CaM]$ ratio decreases and these forms die out. Thus at high [CaM] the effects of Ng are diminished. (E) shows the same effect for physiologically relevant [CaM].

## mCaMKII phosphorylation dose response curve depends nonmonotonically on Ng

Next we looked at the effect of presence or absence of Ng on the phosphorylation of mCaMKII at different [$CaM$]. When we look at phosphorylated mCaMKII (p-mCaMKII) bound to different species of calmodulin, we see that Ng has very different effects on these different species (Fig 3). Phosphorylation by 1- and 2- $Ca^{2+}$/$CaM$ in the presence of Ng is lower than that in the absence of Ng for [$CaM$] $\approx 1 - 100$ μM (Fig 3A, 3B and 3E top 2 panels). As [CaM] increases, the presence of Ng becomes less relevant and the 2 curves converge. The situation is quite different for 3- and more importantly 4- $Ca^{2+}$/$CaM$ species (Fig 3C, 3D and 3E bottom 2 panels). At slightly higher [CaM], the competition with Ng for free CaM plays an important role: a higher proportion of the CaM molecules have 4 calcium ions bound to them in the presence of Ng than in the absence as discussed above, and these CaM molecules give a much higher phosphorylation rate to the CaMKII monomers that they bind. However, these species only exist at relatively lower [CaM]s—as [CaM] increases, the ratio [$Ca^{2+}$]:[$CaM$]$_{free}$ decreases, the effect of Ng becomes insubstantial since the 1- and 2- $Ca^{2+}$/$CaM$ species are responsible for mCaMKII phosphorylation at these concentrations. The dose response curves in the physiologically relevant range of [CaM] ($\sim 10 - 50$ μM, [57–59], Fig 3E) show that the influence of Ng on mCaMKII phosphorylation is most prominent in this range.

The dose response of total mCaMKII phosphorylation to CaM is shown in Fig 4A. Here, the count of p-mCaMKII includes the molecules that have released the $Ca^{2+}$/$CaM$ complexes they were bound to while being phosphorylated. Thus, this count is higher than the total sum of the p-mCaMKII molecules bound to different species of $Ca^{2+}$/$CaM$ plotted in Fig 3. With this information, we can interpret the results shown in Fig 4A after having examined those in Fig 3. Here, the first local maximum is caused by phosphorylation of mCaMKII bound to $4 - Ca^{2+}$/$CaM$. Since there are more of these species in the presence of Ng (Fig 3D), this peak is more prominent when Ng is present in the system. The second peak is caused by phosphorylation of the mCaMKII bound to 1- and 2- $Ca^{2+}$/$CaM$. Finally, at ultra-high [CaM] calcium-free CaM becomes the relevant species, which can still bind CaMKII, albeit with a low affinity (Table 1). Since these species do not allow the phosphorylation of the bound mCaMKII the phosphorylation levels reach 0 at ultra-high [CaM]. These species play an important role at higher [CaM], where the role of Ng is less important. Therefore, total p-mCaMKII increasingly overlaps as [CaM] is increased irrespective of the presence or absence of Ng. The implications of this result can be understood by comparing Fig 4B and 4C, which show the dynamics of mCaMKII phosphorylation with and without Ng at two distinct [CaM]s, an average (30 μM) and an upper bound to physiological (100 μM). The effect of Ng is reversed in the case of these two CaM concentrations: when [$CaM$] = 30 μM, Ng increases mCaMKII phosphorylation level, and when [$CaM$] = 100 μM Ng decreases the overall phosphorylation of CaMKII.

Thus, Ng affects the dose response of mCaMKII phosphorylation to CaM in a nonmonotonic manner and these effects can be clearly understood by examining the effect of this scaffolding molecule on $Ca^{2+}$/$CaM$ species and their ability to activate mCaMKII.

## Ng modulates mCaMKII phosphorylation in response to a $Ca^{2+}$ spike

In dendritic spines, the steady state calcium concentration is $\sim 100 nM$ [60]. Any $Ca^{2+}$ influx is rapidly buffered in the spine, so that [$Ca^{2+}$]$_{free}$ falls back to near steady state levels within $\sim$ 100 ms [61]. To better mimic the experiments, we next used dynamic $Ca^{2+}$ inputs. These inputs were designed such that the free $Ca^{2+}$ ions in the spines reach $\sim 10$ μ$M$ at the peak of the signal and fall back to near steady state levels within $\sim$100 ms [61–66]. We note that this $Ca^{2+}$ is not representative of the actual amount of calcium ions entering the spine, but rather

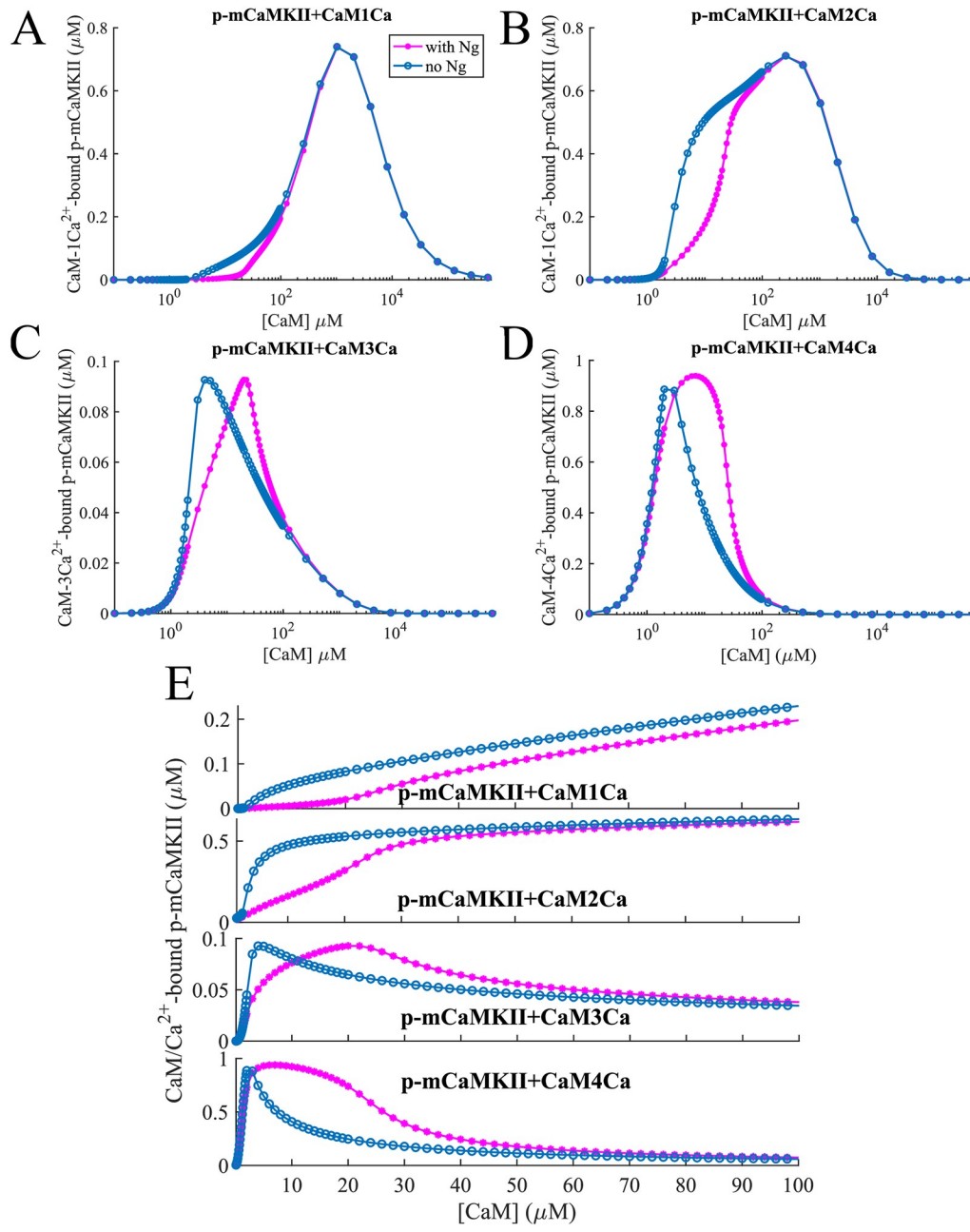

**Fig 3. mCaMKII phosphorylation.** Dose response of phosphorylated mCaMKII molecules bound to (A) 1-, (B) 2-, (C) 3- and (D) 4-calcium states of CaM-$Ca^{2+}$ as shown in Fig 2. At low [CaM], the "multi-calcium-bound" forms of calmodulin play an important role. This is amplified by the fact that these forms when bound to CaMKII give it higher phosphorylation rates (Table 1). (E) sows the same effect for physiologically relevant [CaM]. In these simulations, input $[Ca^{2+}]$ = 10 μM constant.

represents $Ca^{2+}_{free}$. This is because our model does not include any explicit calcium buffers and only simulates calcium buffering mathematically.

Before simulating the calcium spike, we allow the system to equilibrate. There is some basal level of mCaMKII phosphorylation even at the low calcium concentrations ($\sim$ 100 nM) observed in the spine in equilibrium. To achieve a given $[Ca^{2+}]_{free}$ in different conditions, the

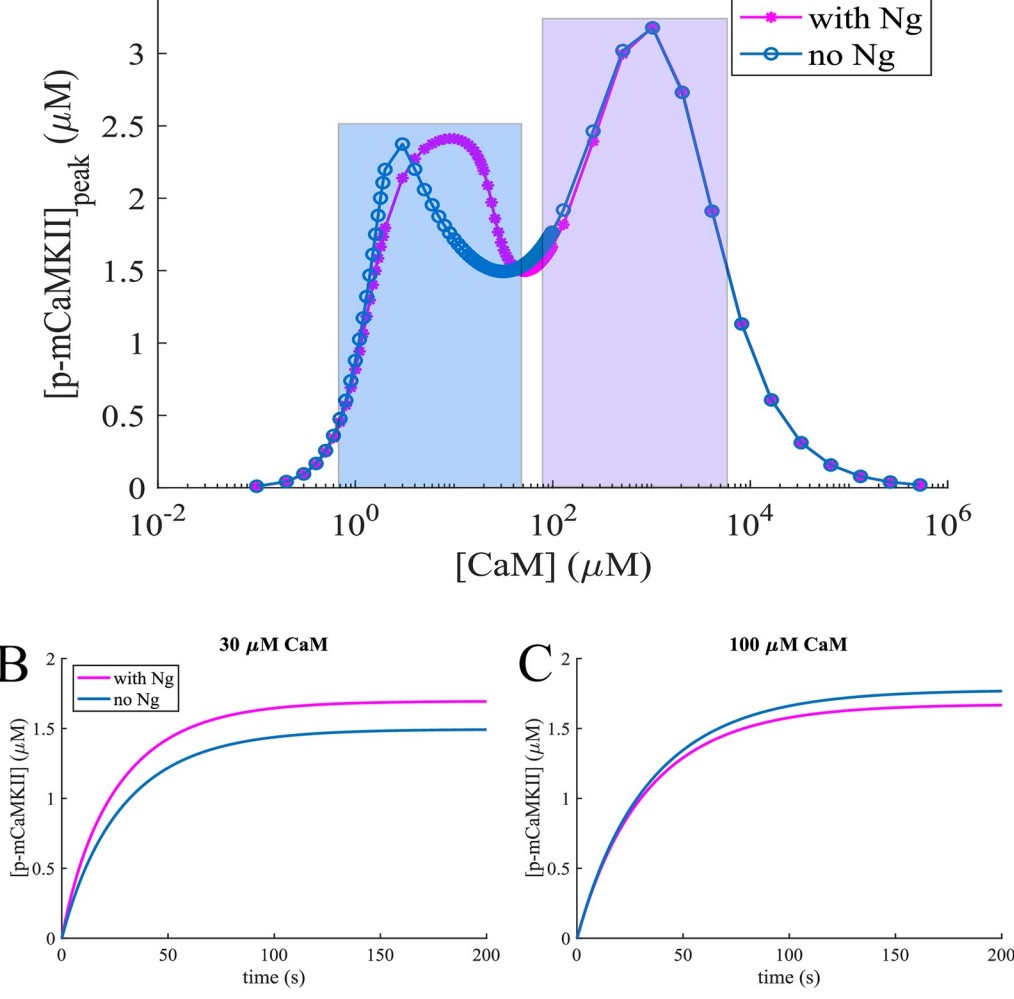

**Fig 4. Phosphorylation dynamics of CaMKII monomers.** mCaMKII phosphorylation dependence on [CaM] and Ng. (A) Steady state phosphorylated mCaMKII concentration as a function of total CaM concentration in the presence of 10 μM $Ca^{2+}$ with (pink lines) and without (blue lines) Ng. The first peak (highlighted in blue rectangle) is caused by CaM molecules bound to 3-4 $Ca^{2+}$ ions, while the second peak is caused (violet rectangle) by CaM molecules bound to 1-2 $Ca^{2+}$s. (B) and (C) Dynamics of mCaMKII (80 μM) autophosphorylation and dephosphorylation by PP1, in the presence of constant $[Ca^{2+}]$ = 10 μM with and without Ng, with $[CaM]$ = 30 μM and $[CaM]$ = 100 μM respectively.

concentration of total calcium varies depending on the conditions simulated, and so does the basal mCaMKII phosphorylation level (S2 Fig). To enable direct comparison, we look at the difference of the peak and basal phosphorylation for each condition, as well as the Area Under the Curve (AUC), peak time, and the decay time of phosphorylation levels.

We first investigated the dependence of maximum mCaMKII phosphorylation on the amplitude of the $[Ca^{2+}]_{free}$ pulse. In Fig 5, the horizontal axis represents the "measured" $[Ca^{2+}]_{free}$ and the vertical axis shows the level of maximal mCaMKII phosphorylation. We note that for any $[Ca^{2+}]_{free}$ the response is higher in the absence of Ng, indicating that even for very small $[Ca^{2+}]$ spikes, the availability of CaM is the limiting factor for mCaMKII phosphorylation. This is further confirmed by the fact that as the size of the $Ca^{2+}$ spike increases, the maximum mCaMKII phosphorylation levels in the presence and absence of Ng diverge from each other as they reach saturation. This observation is true for p-mCaMKII bound to any

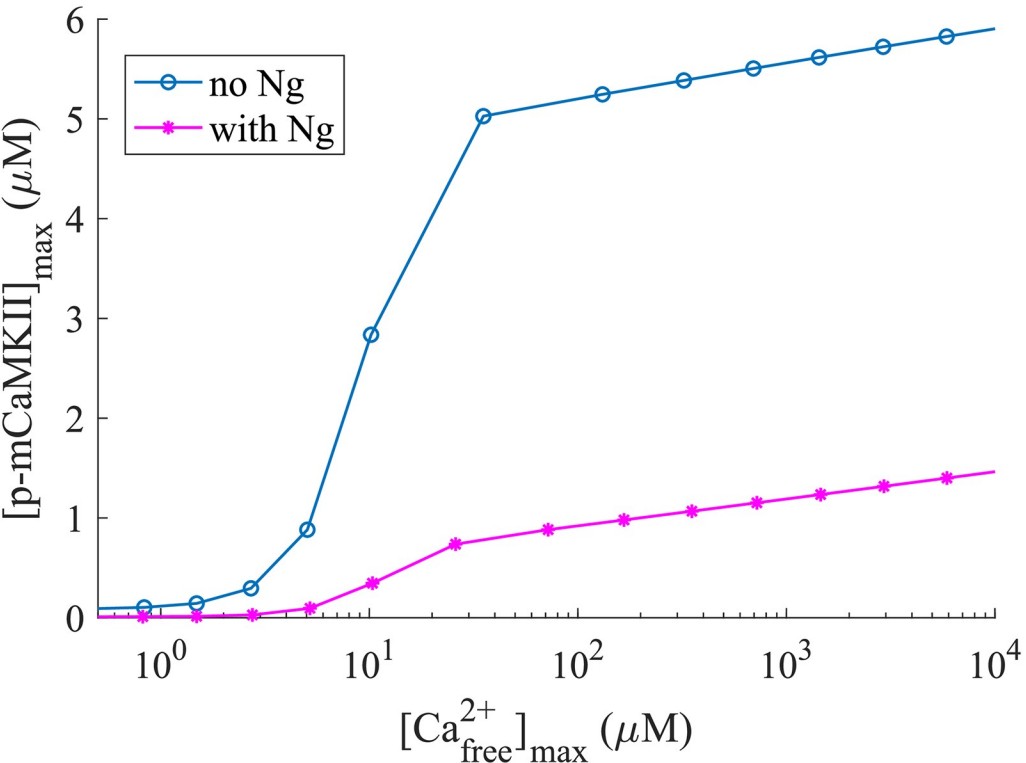

**Fig 5. Maximum phosphorylated monomeric CaMKII concentration for different concentrations of Ca²⁺ spikes with [CaM] = 30 μM.** The horizontal axis represents the free calcium available at the peak of the spike.

species of $Ca^{2+}$/$CaM$, and, as expected, with increasing amplitude of $Ca^{2+}$ spikes, the largest role is played by CaMs that are bound to 4 $Ca^{2+}$s (S3 Fig). This observation, counter-intuitive as it might seem, highlights the importance of the temporal dynamics of the reactions, and demonstrates that drawing conclusions about the system based solely on steady-state considerations alone can be misleading.

The dynamics of mCaMKII phosphorylation is shown in Fig 6A for a range of physiological CaM concentrations including the upper limit of 100 μM. As we can see, even in this extreme case Ng decreases mCaMKII phosphorylation, indicating that [CaM] is the limiting factor. Interestingly, the phosphorylation dynamics with Ng and at 50 μM [CaM] are identical to those without Ng at 30 μM [CaM]. This similarity can be easily understood by looking at the concentrations of Ng-bound CaM (CaM.Ng) at different [CaM]s. As shown in the inset of Fig 6A, when $[CaM] = 10$ μM nearly all of it is bound to Ng, and when $[CaM] > 20$ μM approximately 20 μM of it is bound to Ng.

Further comparing the total amount of mCaMKII phosphorylation over time (Area Under the Curve or AUC) as well as the relative peak phosphorylation levels (Fig 7A and 7C), we see that the presence of Ng makes a striking difference—Ng dramatically impairs phosphorylation of mCaMKII. However, at extreme high [CaM] (100 μM), the system is robust to the presence of Ng although the latter still modulates the level of phosphorylation. The presence of Ng does not appear to have a dramatic effect on the peak time or phosphorylation lifetime, which we define as the time it takes to attain 10% of the peak phosphorylation level (Fig 7E and 7G). This is expected; once phosphorylated, the mCaMKII molecules can be dephosphorylated by PP1, which works independently of Ng and of any other molecule in our model.

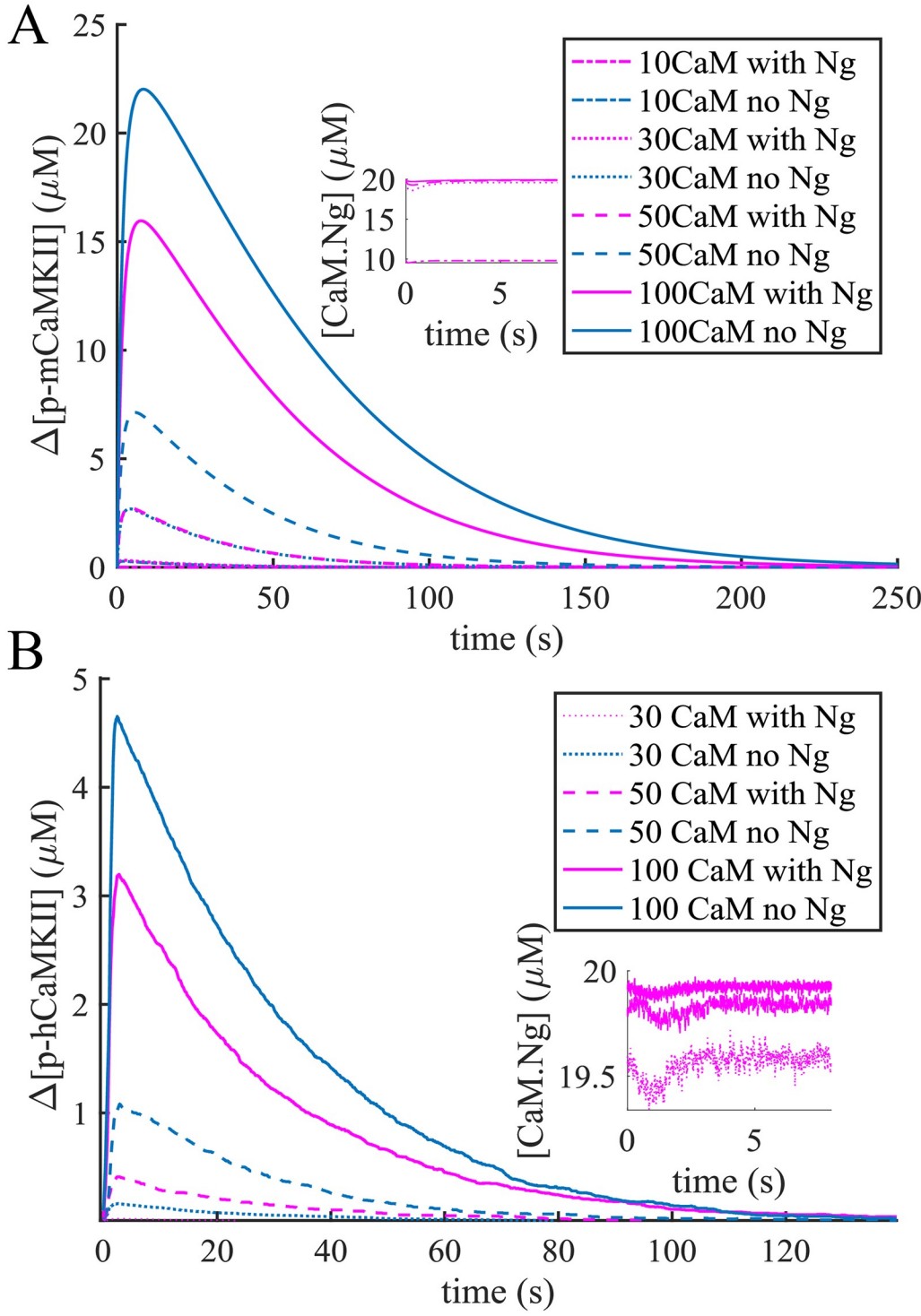

**Fig 6. CaMKII phosphorylation in response to a calcium spike.** The rise in CaMKII phosphorylation level as a response to a $[Ca^{2+}]_{free}$ = 10 μM spike with monomers (A) and holoenzymes (B) for different conditions. The response of the holoenzyme at [CaM] = 10 μM was not significant and is not shown here. The instets show that when $[CaM] > 20$ μM, nearly 20 μM of it is bound to Ng.

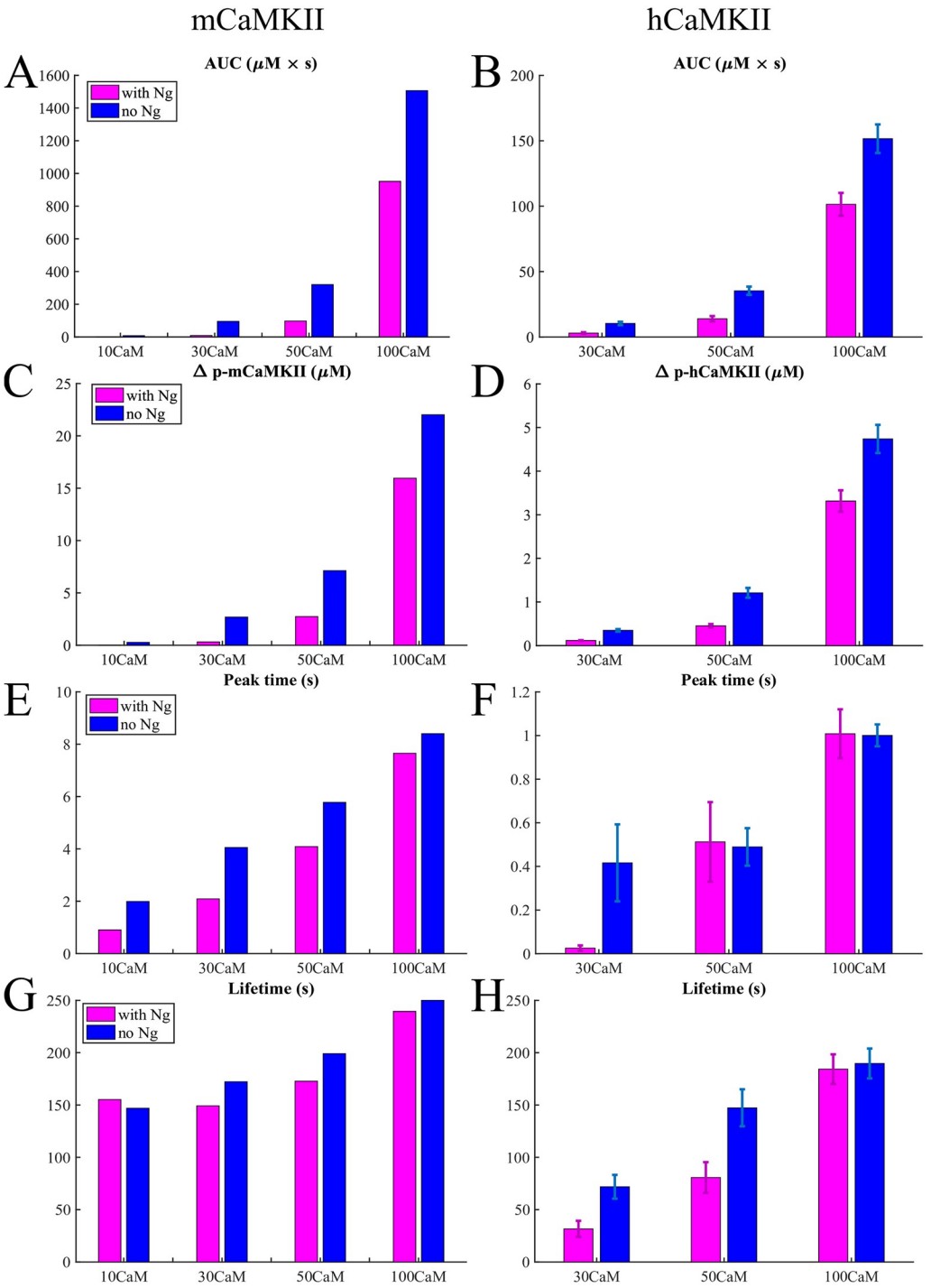

**Fig 7. Quantifying the response to a calcium spike.** Bar graphs in different conditions for the monomers and holoenzymes respectively of Area Under the Curve (AUC) (A) and (B), the maximum increase in phosphorylated CaMKII concentration (C) and (D), the time of the phosphorylation level reaching its peak (E) and (F), and the lifetime defined in the main text (G) and (H) in response to a 10 μM $Ca^{2+}$ pulse.

We note that our simulations result in a phosphorylation lifetime on the order of minutes. This is longer than the time constants of 6 and 45 seconds measured in Ref. [42]. This discrepancy may be because the dephosphorylation time constants measured in spines might be a result of the spatial organization of the spine, not taken into account here, or additional molecular pathways that are not part of the current model.

## CaMKII holoenzyme phosphorylation is robust against fluctuations of $[Ca^{2+}]_{free}$

We next investigated the role of Ng in governing the phosphorylation dynamics of the CaMKII holoenzyme (hCaMKII) using a stochastic agent-based approach. The details of the model development for hCaMKII are given in the Methods. Briefly, the initial conditions of Model 2 contain only CaMKII holoenzymes in the simulated PSD volume $V = 0.0156$ μm$^3$ volume, and the model does not contain any reactions that would allow disintegration of or subunit exchange between holoenzymes. From here onwards, when we refer to CaMKII phosphorylation in the holoenzyme, we are referring to subunits within the holoenzyme that are phosphorylated and we represent this using hCaMKII phosphorylation.

Fig 6B shows the average hCaMKII phosphorylation in response to a ∼10 μM single $Ca^{2+}$ spike for $[CaM]$ = 30 μ$M$, 50 μ$M$, and 100 μ$M$, respectively. We also conducted simulations at $[CaM]$ = 10 μM; however at this CaM concentration, hCaMKII did not react to 10 μM $Ca^{2+}$ pulses. The results shown are an average of ∼30 runs of the stochastic model. In the simulations with $[CaM]$ = 30 μ$M$, the mean peak and standard deviation of $[Ca^{2+}]_{free}$ were 10.7 ± 1.6 μM in the case with no Ng and 10.3 ± 1.2 μM in the case with Ng.

We note that $[CaM]$ = 30 μM corresponds to 283 CaM molecules in our stochastic model. Only a fraction of these molecules binds calcium during the calcium transient, and only a fraction of these complexes bind a hCaMKII subunit. Furthermore, only a fraction of these hCaMKII subunits have an active neighbor that can phsophorylate them. Thus, hCaMKII would not always react to the 10 μM free calcium spike, and sometimes there will be no detected phosphorylated hCaMKII subunits. The events with no detected hCaMKII phosphorylation were not taken into account in the calculations shown in Figs 6B and 7B, 7D, 7F and 7H. In the absence of Ng, 36 out 60 simulations yielded a change in phosphorylation levels. In the presence of Ng, we had to conduct 120 simulations to get 31 simulations that resulted in noticeable hCaMKII phosphorylation. From these observations, we conclude that the probability of hCaMKII phosphorylation is affected by the presence of Ng.

When $[CaM]$ = 50 μM, the mean peak and standard deviation of $[Ca^{2+}]_{free}$ were 9.4 ± 1.2 μM in the case with no Ng and 10.2 ± 1.3 μM in the case with Ng. Finally, in the extreme case when $[CaM]$ = 100 μM the corresponding numbers were 9.2 ± 1.3 μM both with and without Ng. In the inset of Fig 6B we show again that when $[CaM] > 20$ μM nearly 20 μM of it is bound to Ng. In this light, it might be surprising that the response curves for [CaM] = 30, and 50 μM do not overlap as nicely as they do in Fig 6A. This is because the individual simulations are stochastic; comparing the results in Fig 7B, 7D, 7F and 7H we can see that the average results of 30 simulations for [CaM] = 30 μM without Ng, and [CaM] = 50 μM with Ng are consistent with each other within the error bars presented.

By comparing the Figs 6A, 6B and 7A, 7C, 7E and Fig 7B, 7D and 7F, we notice that with the same amount of CaM present in the simulation, the holoenzyme reacts significantly faster to the calcium signal (see the peak time) than the monomeric CaMKII but the phosphorylation level is also much lower. This is understandable: while in the case of monomers any active molecule of CaMKII can bind and phosphorylate another, provided the latter is bound to $Ca^{2+}/CaM$, in the case of the holoenzyme a given subunit can only be phosphorylated by one (and only

one) of its neighbors. Since the phosphorylation of a given subunit in the holoenzyme is still dependent on it being bound to a $[Ca^{2+}/CaM]$ complex, this condition becomes harder to satisfy shortly after the $[Ca^{2+}]$ drops back to $\sim 100$n$M$. And with the holoenzyme the cessation of phosphorylation happens more rapidly, since the conditions imposed for phosphorylation of a given subunit are more rigid. Thus, these more rigid conditions explain both the lower phosphorylation level, and the shorter timescales exhibited by the holoenzyme.

This result tells us that the holoenzyme is more robust to $Ca^{2+}$ fluctuations than the monomers. When the holoenzyme does react to a calcium signal, the phosphorylation lifetime is comparable to that of the monomers (Fig 7G and 7H). And as in the case for monomers, the presence of Ng modulates the effect of the calcium signal at physiological calmodulin concentrations.

## A leaky integrator of $[Ca^{2+}]$ signals

Having established the dynamics of mCaMKII and hCaMKII phosphorylation for a single spike of $Ca^{2+}$, we next investigated the response of hCaMKII to multiple $Ca^{2+}$ spikes. The multiple spikes were designed to mimic experimental stimuli of a train of $Ca^{2+}$ spikes [42, 67–69]. Fig 8 shows the average response of hCaMKII phosphorylation to 10 calcium pulses at 0.5 Hz. As we can see, even with 10 $Ca^{2+}$ pulses in the case of $[CaM] = 30$ μ$M$, the hCaMKII phosphorylation is very small, particularly in the presence of Ng. In fact, even with 10 calcium pulses, only 25 out of 30 simulations resulted in a non-negligible difference of hCaMKII phosphorylation levels in the presence of Ng, and only 28 out of 30 in the absence of Ng. Therefore, multiple spikes increase and even out the probability of hCaMKII phosphorylation but not the overall phosphorylation level in the presence or absence of Ng.

As before, the presence of Ng makes less difference when [CaM] is higher (100 μM), and this system is far more sensitive to calcium signals (Fig 8A, 8C and 8D).

To compare the effect of multiple $Ca^{2+}$ pulses versus a single pulse, we calculated the ratios of the hCaMKII phosphorylation metrics, AUC and maximum change in phosphorylation for 10 to 1 calcium pulses. This ratios are shown in Fig 8B for all 6 conditions tested. As can be seen here, for average physiological conditions, a 10-fold increase in $Ca^{2+}$ signal results in a more modest increase in hCaMKII phosphorylation. With the exception of 100 μM (ultrahigh) CaM concentration, the ratio of the changes in phosphorylated hCaMKII concentrations is consistently lower than 10, and even the ratios of AUCs approach 10 only for [CaM] = 50 μM in the absence of Ng. This result indicates a leakage in the process of the calcium signal integration through hCaMKII phosphorylation. To further investigate the leakiness of hCaMKII phosphorylation, we calculated hCaMKII phosphorylation in response to 30 $Ca^{2+}$ pulses (Fig 9) at average physiological [CaM]. From Fig 9A, we observe that the rate of increase in phosphorylation level decreases as more calcium pulses are added. This leakage of the calcium signal integration is caused by dephosphorylation of hCaMKII by PP1 and has recently been observed experimentally *in vivo* [69].

Since these simulations are conducted stochastically, the timing of $Ca^{2+}$ signals is also stochastic, and not exactly synchronized across the multiple runs. Fig 9B shows hCaMKII phosphorylation in response to a train of 0.5 Hz $Ca^{2+}$ pulses for 3 individual runs. We see that the phosphorylation level rises in a stepwise manner in response to each calcium spike.

To better characterize hCaMKII as a leaky integrator and the effect of Ng, we fit the hCaMKII phosphorylation curves at a physiological $[CaM] = 30$ μM to the leaky integrator equation $x = k - k \cdot e^{-a \cdot t}$, where k is the maximum possible change of phosphorylation level and a is the overall phosphorylation rate (Fig 7C). The curve fitting tool from *MATLAB* was used for this purpose. The response with Ng was characterized by the equation $0.36 - 0.36 \cdot e^{-0.045 \cdot t}$ with an

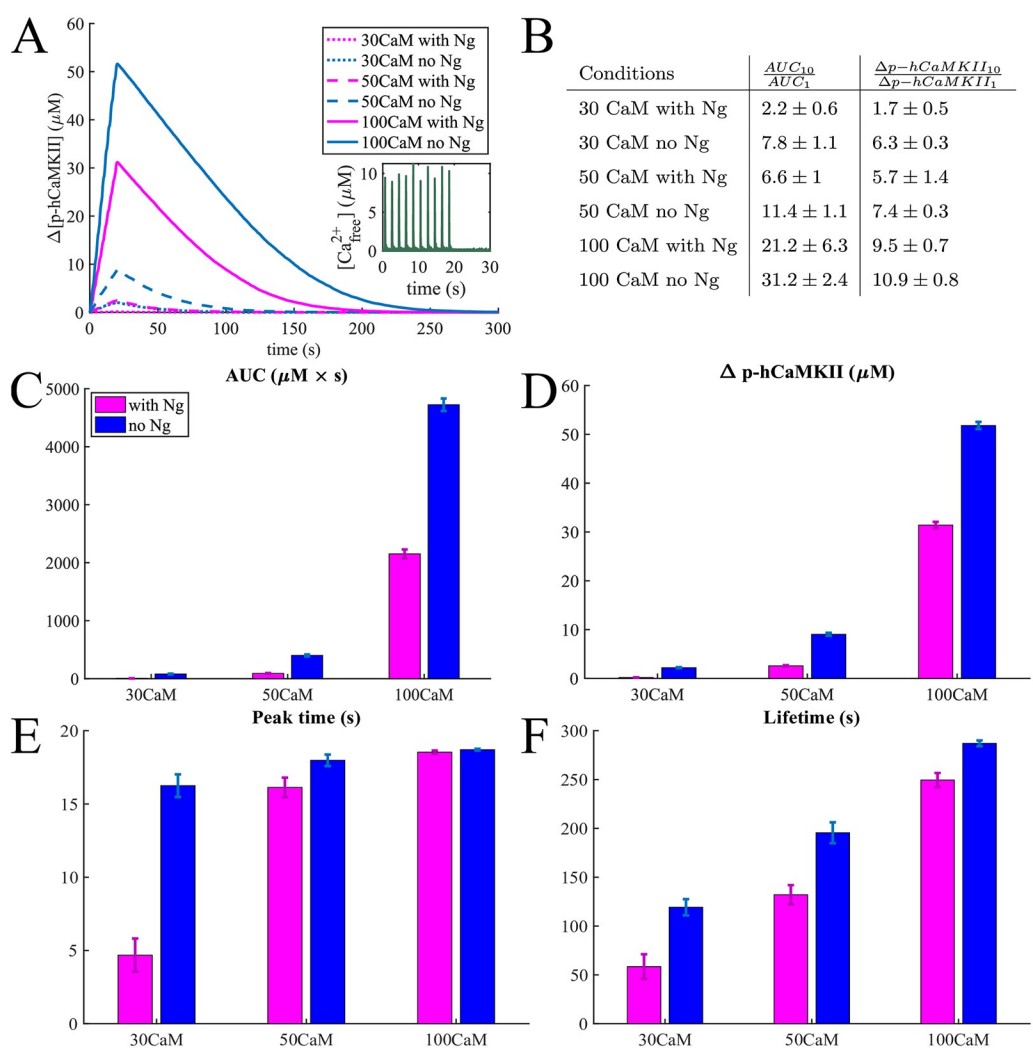

**Fig 8. Responses to trains of 10 $Ca^{2+}$ pulses.** (A) An average phosphorylation of CaMKII holoenzyme in response to 10 $Ca^{2+}$ pulses at 0.5 Hz averaged over 30 simulations with 63 holoenzymes and 30 μM 50 μM, and 100 μM calmodulin. (B) Table of the ratios of the Area under the curve and maximum change in phosphorylation level in response to 10 and 1 $Ca^{2+}$ pulses in different conditions. Bar graphs comparing the (C) Area Under the Curve (AUC), (D) the maximum increase in the hCaMKII phosphorylation concentration, (E) the time of the phosphorylation level reaching its peak, and (F) the lifetime defined in the main text in response these $Ca^{2+}$ pulses.

$R^2$ = 0.9683 and the response without Ng by $7.24 - 7.24 \cdot e^{-0.025 \cdot t}$ with an $R^2$ = 0.9979. This result demonstrates that Ng not only regulates the leak rate of the integrator, but also severely reduces the maximum possible phosphorylation level. The results in Figs 8 and 9 predict that hCaMKII phosphorylation behaves as a leaky integrator of calcium signals, and that the presence of Ng dramatically affects the properties of this integrator.

We also found that hCaMKII activation through $Ca^{2+}/CaM$ binding does not integrate over calcium signals in the same way hCaMKII phosphorylation does. Fig 10 shows 3 representative examples of the changes of CaM-bound hCaMKII concentration in response to $Ca^{2+}$ spikes at $CaM$ = 50 μM and in the presence and absence of Ng. This finding is consistent with recent observations in [44]. We observe that when Ng is present (Fig 10A) hCaMKII activation through CaM binding reaches about the same level at every $Ca^{2+}$ spike. This is what was observed experimentally by [44]. On the other hand, the subsequent decrease of CaM-bound-

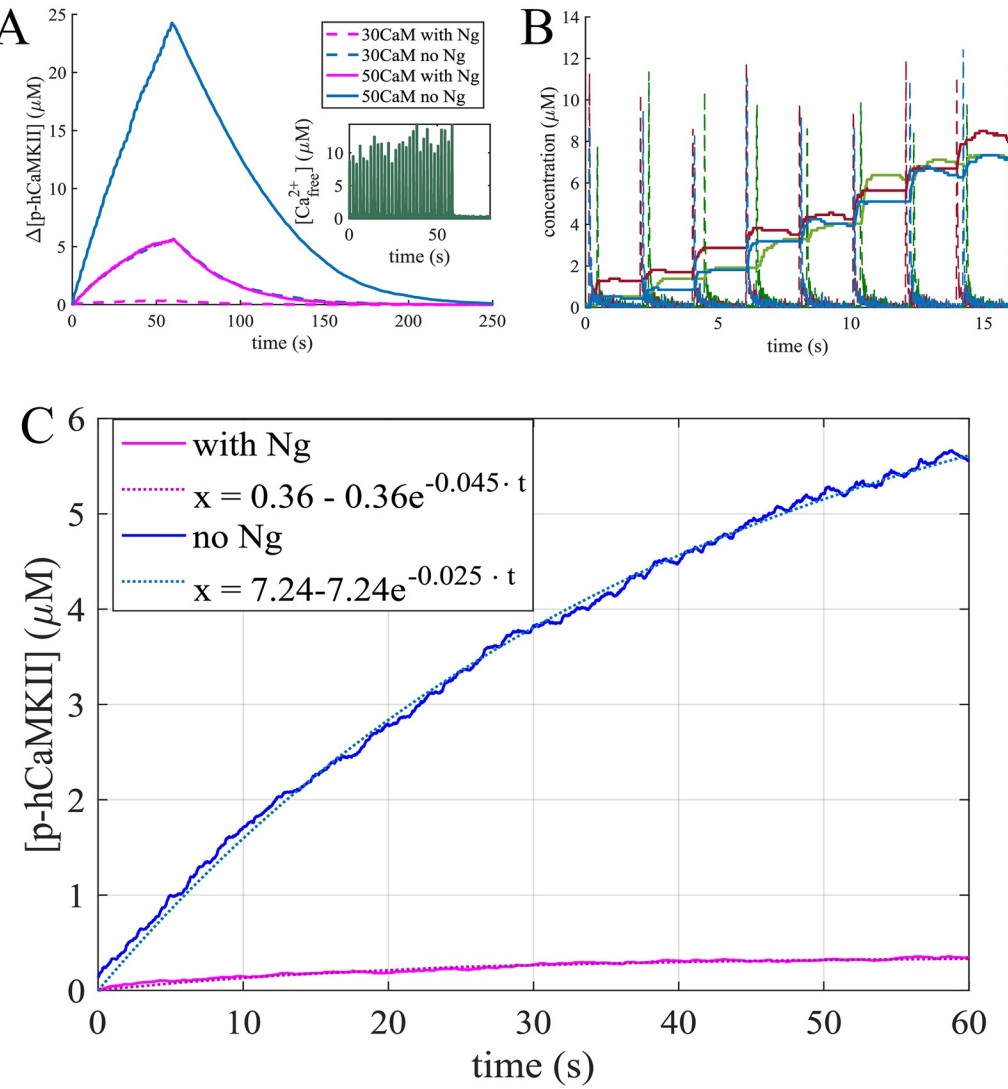

**Fig 9. Responses to trains of 30 Ca²⁺ pulses.** (A) An average phosphorylation of CaMKII holoenzyme in response to 30 $Ca^{2+}$ pulses at 0.5 Hz averaged over 30 simulations with 63 holoenzymes and 30 µM, and 50 µM calmodulin. (B) Three individual simulations: the response of hCaMKII phosphorylation level (solid lines) change to the calcium spikes (dashed lines). (C) Fitting the leaky integrator. The average response to 30 $Ca^{2+}$ pulses was fitted to a curve of the form $x = k - k \cdot e^{-a \cdot t}$ with and without Ng and $[CaM] = 30$ µM, with the curve fitting tool from MATLAB. The parameters obtained are: $k = 7.24$, $a = 0.025$ with an $R^2 = 0.9979$ without Ng and $k = 0.36$, $a = 0.045$ with an $R^2 = 0.9683$ with Ng. Thus Ng not only changes the leaking rate of the integrator, but significantly lowers the maximum possible phosphorylation level.

hCaMKII after each pulse stops at higher concentrations after later $Ca^{2+}$ spikes. This is particularly evident in the absence of Ng (Fig 10B). In fact, in this case, we can also see a slight increase in peak levels of CaM-bound hCaMKII. When we look at only unphosphorylated hCaMKII molecules (u-hCaMKII) bound to CaM, however, we see that the peaks of these curves slightly decrease, while the troughs remain at the same level (S4A and S4B Fig). This prediction was also observed experimentally with T286A CaMKII mutation which make CaMKII unable to be phosphorylated [44]. Finally, when we look at the average peak concentration of CaM bound hCaMKII and u-hCaMKII corresponding to each $Ca^{2+}$ pulse over 30 trials we see that on average peaks corresponding to later pulses are slightly higher for CaM/hCaMKII and slightly lower for CaM/u-hCaMKII species (S4C and S4D Fig). It is noteworthy that, while

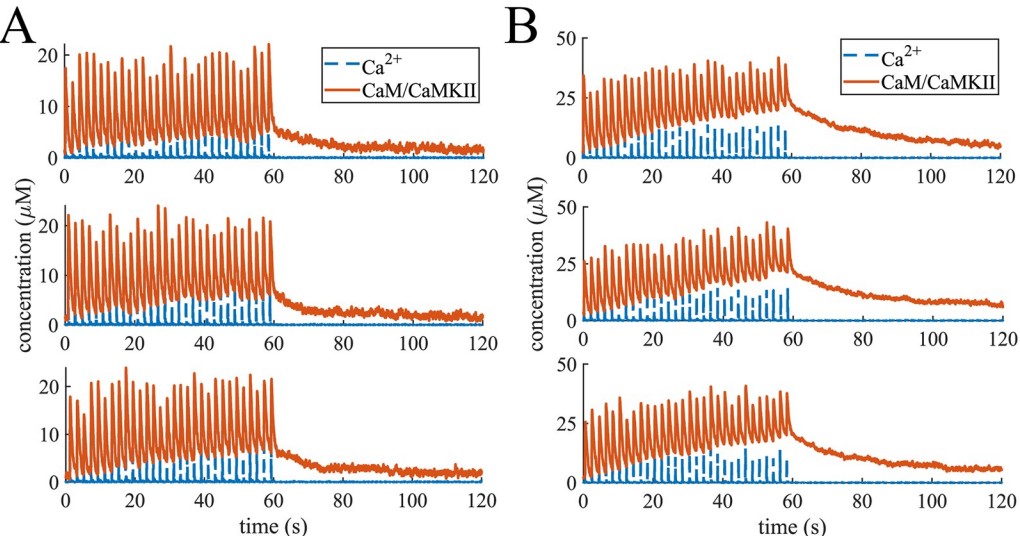

**Fig 10. Responses to trains of 30 Ca²⁺ pulses.** 3 representative curves of CaMKII subunits in the holoenzyme binding CaM in response to $Ca^{2+}$ spikes at $[CaM]$ = 50 μM with (A) and without (B) Ng.

Chang et al [44] observed no increase in CaM/CaMKII peak concentration with later glutamate uncaging pulses, they also found that phosphorylation at T305 site accelerate CaMKII inactivation. The fact that our model does not include this phosphorylation site, might be the reason we see an average increase in peak CaM-bound hCaMKII concentration.

## Discussion

CaMKII phosphorylation is a fundamental process downstream of $Ca^{2+}$-influx in the PSD. There have been many studies focused on the interactions of $Ca^{2+}$, CaM, and CaMKII, and their role in the dynamics of CaMKII phosphorylation [7, 21, 25, 30, 39, 70, 71]. This pathway has also been the focus of computational modeling including explorations of phenomenological models pursuing possibilities of switch-like behavior [11, 33, 35, 37, 38, 40, 72, 73], as well as the effects of intraholoenzyme subunit exchanges on CaMKII activation stability [74], and the role of calmodulin-trapping in maintanance of CaMKII autophosphorylation [39].

In this work, we developed computational models to consider the interactions of both mCaMKII and hCaMKII with CaM in the presence and absence of Ng. These models were built to represent the interactions between different molecules in this pathway based on experimental data with the goal of finely dissecting the behavior of CaMKII phosphorylation. The comparison of our modeling results with the experimental results by Chang et al. [44, 69] can elucidate which aspects of the observed CaMKII behaviour can be explained solely through the interactions of the considered molecules, and which ones can't. We generate the following predictions from our model: *first*, the presence of a scaffolding molecule, Ng, has a nonmonotonic effect on the dose response curve of CaMKII phosphorylation in response to $Ca^{2+}$ influx. *Second*, CaMKII holoenzyme is less sensitive to $Ca^{2+}$ signals than the equivalent number of monomers because of the restrictions in phosphorylation, and is thus less susceptible to noise and fluctuations of $[Ca^{2+}]$. Ng further modulates CaMKII phosphorylation in response to $[Ca^{2+}]$ spikes in the physiological range of $[CaM]$, indicating that $[CaM]$ is the limiting factor for CaMKII activation. It has been observed that CaM becomes a limiting factor for CaMKII activation once regulator of calmodulin signaling (RCS) is phosphorylated, making this protein another competitor to bind CaM [75, 76]. *Finally*, we predict that in the presence of PP1,

the CaMKII holoenzyme acts as a leaky integrator of calcium signals, and Ng significantly affects both the capacity and the leak rate of this integrator (Fig 9C). Based on these findings, we predict that Ng plays a crucial role in fine-tuning the postsynaptic response to $Ca^{2+}$ signals by canceling out the noise and increasing precision in ways that were previously not known.

Our predictions are consistent with experimental data that show that Ng knockout mice exhibit a decrease in LTP induction and spatial learning [77]. Experimental results show that impairment of the binding of Ng to CaM results in lower activity of CaMKII in the dendritic spines [77, 78]. At first sight, this finding seems contradictory to what we find here: the presence of Ng decreases CaMKII activation in our simulations. To understand this apparent inconsistency we note that in our simulations, we compared the results obtained with and without Ng while keeping the overall [CaM] the same for both cases. *In vivo* however, the scaffolding protein Ng sequesters CaM into the dendritic spines, thereby increasing the overall [CaM] in the spines that can be released upon stimulation [79]. This explanation is supported by the observation that postsynaptic injection of $Ca^{2+}/CaM$ enhances the synaptic strength in the same manner as overexpression of neurogranin in CA1 neurons [79–81]. An additional detail, not considered in the present work, is the modulation of Ng activity through protein kinase C (PKC). $Ca^{2+}$ influx in spines activates PKC, which phosphorylates Ng. Phosphorylated Ng has diminished ability to bind CaM [79, 82], resulting in altered dynamics of CaMKII phosphorylation and further fine-tuning of the synaptic response to stimulation. Integrating the detailed pathway of PKC-Ng interaction is the focus of a future study.

Given the wealth of models for CaMKII phosphorylation dynamics in the literature, it is important to consider our results in the context of previous findings. It has been suggested by previous models that CaMKII can exhibit bistability and act as a molecular switch ([35, 37, 38]). While the idea of CaMKII bistability is an appealing explanation behind the LTP phenomenon, it has not been without controversy. For example, one experimental demonstration of CaMKII activation hysteresis was performed *in vitro* at higher than physiological [$Ca^{2+}$] ($> 200 nM$) in the presence of NMDA receptors and in a purified system [83]. To date, to the best of our knowledge there has been no *in vivo* experimental results suggesting evidence for CaMKII bistability. Furthermore, the computational models that predict bistability at physiological $Ca^{2+}$ concentrations require a lower Michaelis-Menten constant of CaMKII dephosphorylation by PP1 ($<1$ μM) than measured experimentally (11 μM, Table 1). A stochastic model of CaMKII activation by Michalski showed that CaMKII does not exhibit bistability in physiological [$Ca^{2+}$] [72] and posited that the previous reports of bistability were perhaps the result of approximations of CaMKII dynamics to overcome the combinatorial explosion of multistate, multisubunit dynamics. Since our model accounts for the subunit level phosphorylation of CaMKII at the holoenzyme level (similar to Michalski), we did not find regimes of bistability with our model. Indeed, even if we initiated the simulation with all of the hCaMKII in the model being phosphorylated, the activation decays back to basic level at physiological [$Ca^{2+}$] = 100 nM (S4 Fig).

The role of Ng in modulating the dynamics of CaMKII, particularly, the leak rate and capacity of CaMKII phosphorylation, has multiple implications in downstream processes in a spine. Phosphorylation of CaMKII activates its kinase domain, which leads to subsequent modification of the dendritic spine and the postsynaptic density [42, 84–86]. Specifically phosphorylated CaMKII unbinds from F-actin in the spine allowing reorganization of the cytoskeleton, and rebinds this newly structured F-actin after it has been dephosphorylated [16, 84, 87–90]. These interactions with the actin cytoskeleton allow for dynamic rearrangements of the actin organization within the spine, subsequently impacting its size and shape [91–93] (Fig 11). CaMKII activation also leads to increased trafficking [94, 95] and trapping [96] of AMPA receptors at the PSD as well as an increase in the conductance of these receptors after the

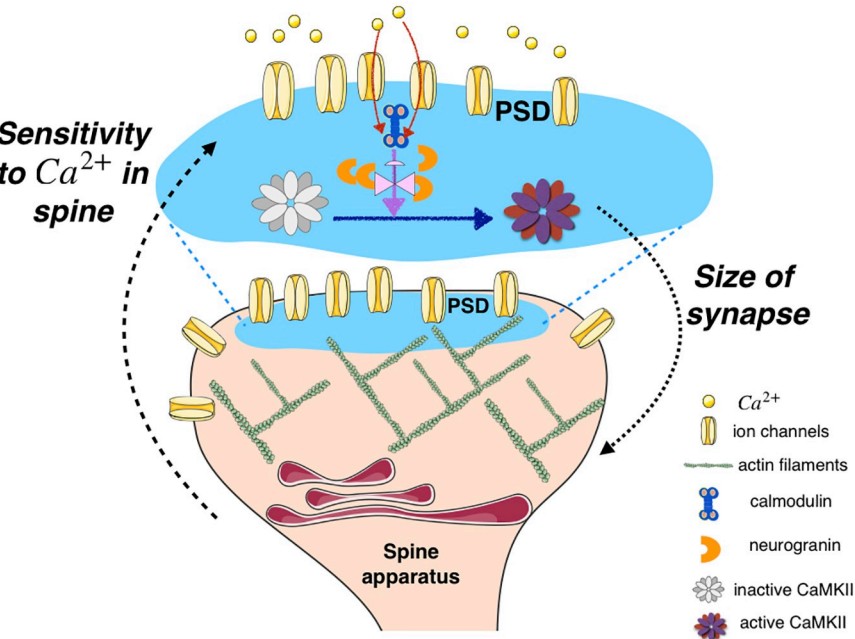

**Fig 11. Schematic representation of CaMKII phosphorylation controlling sensitivity of a synapse to $Ca^{2+}$ signals.**
In the PSD, influx of $Ca^{2+}$ through the ion channel, binding of $Ca^{2+}$ to calmodulin, and phosphorylation of CaMKII
sets off the series of events associated with synaptic plasticity. In this work, we show that the competition between
neurogranin sequestration of calmodulin and $Ca^{2+}$ binding of calmodulin affects CaMKII dynamics. This has
implications for the feedback between $Ca^{2+}$ in the spine and structural plasticity of the spine, particularly size
dynamics.

induction of LTP [97–100]. All of these events result in changes to the size and strength of the
synapse [36, 101–103] (Fig 11).

Synaptic size and strength are regulated by firing history [104] with very high precision [1].
As a synapse undergoes increased LTP, it grows larger and subsequently the $Ca^{2+}$ transients in
the spine become increasingly dilute in response to back-propagating action potentials [65].
Thus, the integration of $Ca^{2+}$ signals is a built-in mechanism in the dendritic spine to down-
regulate its sensitivity to these signals as the corresponding synapse grows larger (Fig 11). Our
study shows that scaffolding molecules such as Ng can modulate these dynamics. The results
of our simulations combined with experimental findings cited above point to an important
role for Ng in regulating the postsynaptic response to $Ca^{2+}$ signals. This finding adds to the
growing evidence that scaffolding proteins fine-tune the signaling pathways in biological signal
transduction mechanisms [105–107].

The model described here will enable the development of more detailed models of signaling
events in the PSD including coupling between CaMKII, PKC, and PKA [74, 82, 108, 109], spa-
tial organization of calcium-CaM-CaMKII dynamics building on [1, 65, 110–112], and coupling
between CaMKII and actin interactions [92, 113]. Such efforts will be necessary to understand
the mechanochemical coupling within spines and how they regulate information processing.

## Supporting information

**S1 Fig. Interaction of $Ca^{2+}$ with CaM in the absence of CaMKII.** Relative distribusion of
$Ca^{2+}$-bound CaM species in the absence of CaMKII.
(TIF)

**S2 Fig. Baseline levels of mCaMKII phosphorylation.** (A) at 100 nM $[Ca^{2+}]$ different concentrations of CaM result in different baseline levels of mCaMKII phosphorylation. (B) to enable comparisons between different conditions we compare the increase in mCaMKII phosphorylation levels ($\Delta$[p-mCaMKII]) as shown.
(TIF)

**S3 Fig. Dose response to $Ca^{2+}$ spikes.** Dose response to $Ca^{2+}$ spikes factored into p-mCaMKII bound to different species of $Ca^{2+}$/$CaM$.
(TIF)

**S4 Fig. CaM-bound CaMKII.** Comparing total and unphosphorylated hCaMKII (u-hCaMKII) bound to CaM with (A) and without (B) Ng: while total CaM-bound hCaMKII decays slower after a $Ca^{2+}$ spike, CaM-bound u-hCaMKII gets down to base level immediately after the spike. C and D show the 30 peaks reached by CaM-bound hCaMKII and u-hCaMKII averaged over 30 simulations, with and without Ng respectively.
(TIF)

**S5 Fig. Absence of bistability.** hCaMKII does not exhibit bistability at $[Ca^{2+}] = 100 nM$ in our model: even when all of hCaMKII is phosphorylated, the activation decays back to base level.
(TIF)

## Acknowledgments

We thank Miriam Bell for useful discussions and assistance with making Fig 11. We thank Miriam Bell, Allen Leung and Kiersten Scott for proofreading the manuscript.

## Author Contributions

**Conceptualization:** Mariam Ordyan, Tom Bartol, Mary Kennedy, Padmini Rangamani, Terrence Sejnowski.

**Data curation:** Mariam Ordyan.

**Formal analysis:** Mariam Ordyan.

**Funding acquisition:** Tom Bartol, Mary Kennedy, Padmini Rangamani, Terrence Sejnowski.

**Investigation:** Mariam Ordyan, Tom Bartol, Mary Kennedy, Padmini Rangamani, Terrence Sejnowski.

**Methodology:** Mariam Ordyan, Tom Bartol, Padmini Rangamani, Terrence Sejnowski.

**Project administration:** Padmini Rangamani, Terrence Sejnowski.

**Software:** Mariam Ordyan, Tom Bartol.

**Supervision:** Padmini Rangamani, Terrence Sejnowski.

**Visualization:** Mariam Ordyan.

**Writing – original draft:** Mariam Ordyan, Padmini Rangamani.

**Writing – review & editing:** Mariam Ordyan, Tom Bartol, Mary Kennedy, Padmini Rangamani, Terrence Sejnowski.

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
