## [Decision Letter · Decision Letter 0]

4 Feb 2020

Dear Dr. Rangamani,

Thank you very much for submitting your manuscript "Interactions between calmodulin and neurogranin govern the dynamics of CaMKII as a leaky integrator" for consideration at PLOS Computational Biology.

As with all papers reviewed by the journal, your manuscript was reviewed by members of the editorial board and by several independent reviewers. In light of the reviews (below this email), we would like to invite the resubmission of a significantly-revised version that takes into account the reviewers' comments.

We cannot make any decision about publication until we have seen the revised manuscript and your response to the reviewers' comments. Your revised manuscript is also likely to be sent to reviewers for further evaluation.

Sincerely,

Joanna Jędrzejewska-Szmek, Ph.D.

Guest Editor

PLOS Computational Biology

Kim Blackwell

Deputy Editor

PLOS Computational Biology

Reviewer's Responses to Questions

**Comments to the Authors:**

Reviewer #1: This manuscript from Ordyan et al., developed a new model of CaMKII activation by Ca2+ influx into dendritic spines based on a rule-based modeling. Since CaMKII is a kinase important for LTP, learning and memory, the detailed understanding of CaMKII activation mechanism would be important for understanding the biochemical process occurring in spines during LTP. In this manuscript, the authors attempted to model the effect of neurogranin, a molecule known to cache Ca2+-free calmodulin (CaM) but not Ca2+ CaM. Thus, neurogranin can potentially modulate CaMKII. While there were several attempt to develop a model for CaMKII activation, the molecular interactions including CaMKII, neurogranin, phosphatase and CaM are complicated and it was difficult to simulate. They adapted a method using rule-based modeling to perform this simulation. Overall the manuscript shows important information about signal transduction in spines.

However, there are several limitations in the manuscript in the current form:

1. Previously several different models of CaMKII activation in spines have been developed. One of the most relevant ones would be a model developed by Zhabotinsky et al (2002, J. Neurosci), as it includes all molecular components discussed in the current paper, including CaM, CaMKII, Neurogranin and phosphatase. The authors should go through previous models of CaM-neurogranin binding and its role in CaMKII activation in the introduction and discussion. In particular, the paper would be enhanced by more explicit explanation of the difference from previous attempts and benefits of the presented model.

2. Dephosphorylation rate of bulk CaMKII in spines was measured to be ~6 s (Lee et al., 2009 Nature; Chang et al., 2017 Neuron, Chang et al., 2019; Nat Commu) at room temperature and perhaps much faster in physiological temperature, instead of ~60 s used in the simulation. The author could comment on the impact of this on the simulation.

3. The paper would benefit from additional discussion about the kinetics of CaM-CaMKII binding in the simulation compared with that measured in spines (Chang et al., 2019, Nat Commu).

4. Since Neurogranin is phosphorylated by PKC, which is activated by Ca2+ influx into spines (Colgan et al., 2018, Nat Neuro), and releases CaM as suggested by Zhabotinsky et al. (2002, J. Neurosci). Discussion on potential impact to their simulation would enhance the paper.

Reviewer #2: The manuscript by Ordyan et al addresses an important question around how CaMKII activation is controlled in spines. The model is based on earlier models and is an as solid model as one could wish for to make predictions and increase our understanding for the dynamic signaling underlying CamKII dependent plasticity in spines. The simulations are testing the role of Cam, Ng and Ca for the activation of CaMKII (incl phosphorylation). Several important predictions are made, and to increase the understanding from the readers point of view, additional explanations will be useful. Since the model is not that big and one can measure all variables in parallel and manipulate amounts, kinetics, etc, one should also be able to ‘explain’ the mechanisms in the model that the results depend on. Thus ideally the results are not only model predictions, but ‘explainable’ predictions given the current model. I have listed questions I asked myself when reading the manuscript. If these items can be explained in even more intuitive ways, this work will make a much larger impact.

a) Explain better (e.g. on p6 or in methods) how the Ca input was modelled. And maybe show the input signal (it is shown in fig 8, but would be good to see one of the 10mikroM transients used for the first few figs in a clearer way). Was the Ca_free level ‘clamped’ to a certain level, e.g. during the 10mikroM transients, such that it didn’t matter how much Cam was around? Thus much more Ca was bound to Cam in those simulations where the Cam amount was higher (thus assuming that more calcium was sequestered in the PSD)? If one instead had ‘injected’ a certain amount of Ca for each input (and assumed a ‘removal machinery’ for the Ca such that it went down to resting state within 100ms or so), how would that have affected the results? Then perhaps one shouldn’t necessarily see an increased CamKII activation for more Cam. Please discuss (or test) these things and relate to a reasonable scenario for real spines. Perhaps one could speculate that if there were more Cam it also requires higher Ca influx for efficient CamKII activation? Is also calcium limiting if Cam is there in high amounts, and vice versa?

b) On p7 it is said that “Interestingly, the phosphorylation dynamics with Ng and a 50 mikroM Cam is identical to that without Ng at 30 mikroM Cam”. Can this be explained? Is this because 20 mikroM Ng binds almost 20 mikroM Cam, thus 30mikroM available to bind to Ca? But then, why is this not the case for the holoenzyme in Fig 6B?

c) In Fig 2, the presence of Ng can allow a higher relative Cam4Ca amount. This is explained with the Ca/Cam ratio. Is this behavior also seen if one assumes the CamKII amount is zero? I.e. is this behaviour a result of only competition between Ca and Ng for Cam, or does CamKII play a role as a ‘leaky buffer’ for the multi-calcium-bound forms?

d) In fig 4A the steady state activation of CamKII is shown for higher Cam levels. The second peak is due to Cam with few Ca bound. Is the reason that we don’t see that for the transient Ca input used in fig 3 that there is not enough time for that during 100ms or so? One can perhaps also notice that for physiological levels of Cam (e.g. 10mikroM) the response reaches up to 2.5 mikroM. If one compares with fig 3 the amplitude is just around 1. It thus seems that during a Ca transient less than half of the steady-state level is reached. But in fig 4C and 4D it seems to take a long time to reach steady-state. Explain these things better in the text, please.

e) Fig 5, how can pCamKII_max be so much higher without Ng? What is the relation between fig 5 and earlier figs? Also in earlier figs the results for 30 mikroM of Cam is shown at a Ca transient of 10 mikroM. Describe better and explain so readers can understand. Is the input changed here?

f) Fig 6, why is the holoenzyme less activated than the monomer system? Is that because only the closest neighbours can be phosphorylated in the holoenzyme? Also explain what mechanism in the model makes the holoenzyme both peak earlier and decay more rapidly.

g) The manuscript highlights that sequestration effects are important, and that Cam becomes a limiting species. In this context one I think should mention another computational study that reached a similar conclusion that Cam is limiting for activating CamKII if one has phosphorylated RCS (ARPP-21), see Nair et al, Plos Comput Biol, 2016. Also Rakhilin et al 2004 is in line with that Cam limiting even for PP2B activation and that RCS competes with PP2B for activated Cam.

h) Since Cam binding to Ng is highlighted here and there, why not show an inset plot of the Ng-Cam in relevant figures (e.g. see also item b) above).

i) Please discuss the role PKC phosphorylation of Ng might have, and relate to the figs in the manuscript. For example, if using more Cam in fig 3D or 4A it would not help much, but if one has more Ca it might change things? Or how does the model behave here?

j) In the methods section I think one should just acknowledge that rule-based modeling in this case implies that one assumes that allosteric effects are not so important. (although obviously it would be unpractical to simulate all combinations of molecular species, and one would not have data on allosteric interaction anyway, at least not today).

k) Finally, maybe make sure fig texts are more transparent when it comes to what Ca input that was used, and also if monomers or holoenzymes were the readout. Seems monomers were used in the beginning and then holoenzymes investigated.

Reviewer #3: Interactions between calmodulin and neurogranin govern the dynamics of CaMKII as leaky integrator

Mariam Ordyan, Tom Bartol, Mary Kennedy, Padmini Rangamani, and Terrence Sejnowski

In this manuscript, the authors investigate the phosphorylation dynamics of CaMKII triggered through binding of calcium-associated calmodulin. The authors account for the sequestering of calmodulin through neurogranin and show that the presence of neurogranin has a profound impact on available calmodulin and consequently on CaMKII phosphorylation levels and dynamics. The results suggest neurogranin to be potentially important for regulating processes involved in synaptic long-term potentiation. The authors are the first to consider the interaction between calmodulin and neurogranin and the impact for CaMKII activation.

My general critic is that the study falls short in providing the insights and explanations which are the unique potential of such modeling studies. Another crucial part of biophysical modeling work is to propose feasible experiments which can support or contradict the conclusions drawn from the theoretical results and further elucidate the role of the protein interaction network. Such considerations are completely missing in the present work.

I have a couple of main comments and concerns :

1. It's also not clear how the steady-state results, which reveal the “non-intuitive” dependence of pCaMKII on calmodulin concentration (e.g. Fig. 4), relate to the dynamic stimulation part, in which the authors use calcium transients to induce CaMKII phosphorylation (starting with Fig. 5). The facilitating effect of neurogranin in the steady-state results seems to be absent in the dynamic calcium simulations, where the sole effect of Ng appears to be reducing pCaMKII levels. Putting the presented results in relation would help reader comprehension.

2. As the authors point out, calmodulin contains four calcium binding sites, two at the C- and two at the N-terminal domain. One hallmark of a steep calcium-dependent activation of calmodulin is that calcium binding happens in a cooperative manner in each one of these pairs (Chin D, Means A (2000) Calmodulin: a prototypical calcium sensor. Trends Cell Biol 10: 322–328). It's surprising that the authors don't account for this property crucially determining how calcium activation affects downstream targets such as CaMKII. The cooperativity strongly favors CaM+2Ca and CaM-4Ca alternating the relative concentrations curves in Fig. 2 and all subsequent results.

3. Previous work on CaMKII activation through CaM/Ca and autophosphorylation revealed that the protein can exhibit bistability in its phosphorylation level (Zhabotinsky 2000 Biophys J; Graupner 2007 PLoS Comp Biol). In other words, for the same calcium concentration, the protein can exist in a highly or a weakly phosphorylated state and both are stable. How does this current work relate to this line of work. Does CaMKII bistability exist in the model?

4. What is the rational behind depicting relative concentration for ca-bound calmodulin in Fig. 2?

5. Fig. 3 and Fig. 4 : Why is the concentration of calcium-bound calmodulin bound to phosporylated CaMKII (Fig. 3) and phosphorylated CaMKII (Fig. 4) decreasing with increasing calmodulin concentration? Even though the relative free calcium-bound calmodulin decreases with more CaM, the absolute concentration of calcium-bound calmodulin should increase or saturate. I would expect monotonously increasing concentration levels in Fig. 3 and 4.

6. pg. 4, 3rd paragraph : What is at the origin of the cross-over of the of the calcium-bound calmodulin with multiple calcium ions between curves in the presence and absence of Ng? Can the authors provide an intuition for this effect which does not exist for calmodulin bound with 2 or 3 calcium ions.

7. It would be instructive to see the dynamics of the different calmodulin forms (CaM-1Ca, CaM-2Ca,...) in response to the calcium transient, the summary of which is presented in Fig. 5 for pCaMKII. That could maybe also provide a link to the steady-state considerations discussed up to this point.

8. Until Figure 4, the authors emphasize that there exists a calmodulin concentration (~30 μM) for which the presence of Ng favors CaMKII phosphorylation. However, this facilitation seems to be gone when simulation calcium transients (Fig. 5-9). What is the reason for this?

9. pg. 14. 1st sentence : “We next investigated ... “. Which approach was used for the results until that point? What are the differences in the approaches and how to interpret the results from both? Why are the results quantitatively different in terms of peak phosphorylated CaMKII concentration, for example (Fig. 6)?

10. pg. 14. 1st par, line 12 : It is not clear to me why some of the simulations would not yield a change in CaMKII phosphorylation level and other yield a considerable increase in pCaMKII. Can the authors elaborate and explain?

Minor comment :

Fig. 2 A-D : I would suggest the same x-scale for all four panels. Same for Fig. 3A-C.

**Have all data underlying the figures and results presented in the manuscript been provided?**

Reviewer #1: Yes

Reviewer #2: Yes

Reviewer #3: None

PLOS authors have the option to publish the peer review history of their article (what does this mean?). If published, this will include your full peer review and any attached files.

Reviewer #1: No

Reviewer #2: No

Reviewer #3: No
---

## [Decision Letter · Decision Letter 1]

20 May 2020

Dear Dr. Rangamani,

Thank you very much for submitting your manuscript "Interactions between calmodulin and neurogranin govern the dynamics of CaMKII as a leaky integrator" for consideration at PLOS Computational Biology. As with all papers reviewed by the journal, your manuscript was reviewed by members of the editorial board and by several independent reviewers. The reviewers appreciated the attention to an important topic. Based on the reviews, we are likely to accept this manuscript for publication, providing that you modify the manuscript according to the review recommendations.

Sincerely,

Joanna Jędrzejewska-Szmek, Ph.D.

Guest Editor

PLOS Computational Biology

Kim Blackwell

Deputy Editor

PLOS Computational Biology

[LINK]

Reviewer's Responses to Questions

**Comments to the Authors:**

Reviewer #1: The authors addressed all concerns from this review.

Reviewer #2: The authors have addressed all my previous comments and questions.

Reviewer #3: the comments are uploaded as attachment

**Have all data underlying the figures and results presented in the manuscript been provided?**

Reviewer #1: Yes

Reviewer #2: None

Reviewer #3: Yes

PLOS authors have the option to publish the peer review history of their article (what does this mean?). If published, this will include your full peer review and any attached files.

Reviewer #1: No

Reviewer #2: No

Reviewer #3: No
---

## [Decision Letter · Decision Letter 2]

4 Jun 2020

Dear Dr. Rangamani,

We are pleased to inform you that your manuscript 'Interactions between calmodulin and neurogranin govern the dynamics of CaMKII as a leaky integrator' has been provisionally accepted for publication in PLOS Computational Biology.

Best regards,

Joanna Jędrzejewska-Szmek, Ph.D.

Guest Editor

PLOS Computational Biology

Kim Blackwell

Deputy Editor

PLOS Computational Biology

Reviewer's Responses to Questions

**Comments to the Authors:**

Reviewer #3: The authors have addressed the issues raised.

I simply would like to add a comment regarding this exchange :

Me : I understand that the model parameters are outside the range where CaMKII would exhibit bistable phos-

phorylation behavior. However, the presentation of the authors suggest that the difference between previous

models and their findings is the nonlinear rate functions used in the model (pg. 4, 1st line) and the fact

that they use a more complete computational model of CaMKII dynamics [accounting] for both the behavior

of the monomer and the dynamics of CaMKII holoenzyme. Both are not differences to previous studies

mentioned above, which resolve the nature of the inter-subunit phosphorylation in the holoenzyme. Also,

the nonlinear rate functions emerge from the non-linear calcium-dependent activation of CaMKII phospho-

rylation and dephosphorylation (often described by Hill functions). I suspect that the same behavior exists

in the presented model?

Authors : Unfortunately, we were quite puzzled by the reviewer’s reference to nonlinear rate functions (pg 4., 1st

line). We are not sure what the reviewer is referring to here. However, we seek to clarify that we do not

approximate the reaction rates through Hill functions, rather all the rate constants used in our model are

constant values and rate equations are linear fluxes, as presented in Table 1. The non-linearity of calcium-

dependent activation of CaMKII in our model arises from the cooperativity built into the rate constants

and reactions. To prevent any further confusion we have now added a list of the reactions used to our

supplemental materials (Table 2). As noted in our previous response, based on the experimentally measured

parameters used in our model we do not expect nor observe bistability (Figure 4S).

Me : The authors refer in their introduction themselves to nonlinear rate functions (pg. 4., 1st line) as used in previous modeling studies.

My point is : the presentation of the differences between the study presented in the manuscript and previous modeling efforts is not correctly presented in this paragraph of the introduction (last par. pg. 3 - first par. pg. 4).

- Previous studies resolve the nature of the inter-subunit phosphorylation in the holoenzyme.

- And, as the authors correctly point out, the cooperative nature of calcium activation leads to non-linear activation and dephosphorylation of the CaMKII protein.

These are not the differences which explain that the authors do not observe bistability. Simply, the model operates in a parameter regime in which the bistable behavior does not occur.

**Have all data underlying the figures and results presented in the manuscript been provided?**

Reviewer #3: None

PLOS authors have the option to publish the peer review history of their article (what does this mean?). If published, this will include your full peer review and any attached files.

Reviewer #3: No

---

## [Editor Report · Acceptance letter]

10 Jul 2020

PCOMPBIOL-D-19-02155R2 

Interactions between calmodulin and neurogranin govern the dynamics of CaMKII as a leaky integrator

Dear Dr Rangamani,

I am pleased to inform you that your manuscript has been formally accepted for publication in PLOS Computational Biology. Your manuscript is now with our production department and you will be notified of the publication date in due course.

With kind regards,

Laura Mallard
